# Anti-ferroptotic mechanism of IL4i1-mediated amino acid metabolism

Leonie Zeitler[1†], Alessandra Fiore[1†], Claudia Meyer[2], Marion Russier[1], Gaia Zanella[1], Sabine Suppmann[1], Marco Gargaro[3], Sachdev S Sidhu[4], Somasekar Seshagiri[5], Caspar Ohnmacht[6], Thomas Köcher[7], Francesca Fallarino[3], Andreas Linkermann[2], Peter J Murray[1]*

[1]Max Planck Institute of Biochemistry, Martinsried, Germany; [2]Universitätsklinikum Carl Gustav Carus Dresden, Dresden, Germany; [3]Università degli Studi di Perugia, Perugia, Italy; [4]The Donnelly Centre for Cellular and Biomolecular Research, University of Toronto, Toronto, Canada; [5]SciGenom Research Foundation, Bangalore, India; [6]Helmholtz Zentrum München Center of Allergy and Environment (ZAUM), Technical University and Helmholtz Center Munich, Munich, Germany; [7]Vienna BioCenter Core Facilities GmbH, Vienna, Austria

**Abstract** Interleukin-4-induced-1 (IL4i1) is an amino acid oxidase secreted from immune cells. Recent observations have suggested that IL4i1 is pro-tumorigenic via unknown mechanisms. As IL4i1 has homologs in snake venoms (L-amino acid oxidases [LAAO]), we used comparative approaches to gain insight into the mechanistic basis of how conserved amino acid oxidases regulate cell fate and function. Using mammalian expressed recombinant proteins, we found that venom LAAO kills cells via hydrogen peroxide generation. By contrast, mammalian IL4i1 is non-cytotoxic and instead elicits a cell protective gene expression program inhibiting ferroptotic redox death by generating indole-3-pyruvate (I3P) from tryptophan. I3P suppresses ferroptosis by direct free radical scavenging and through the activation of an anti-oxidative gene expression program. Thus, the pro-tumor effects of IL4i1 are likely mediated by local anti-ferroptotic pathways via aromatic amino acid metabolism, arguing that an IL4i1 inhibitor may modulate tumor cell death pathways.

*For correspondence:
murray@biochem.mpg.de

†These authors contributed equally to this work

## Introduction

Most cells of the immune system depend on external supplies of all essential and most non-essential amino acids for proliferation in active immune responses, and for diverse effector functions, such as nitric oxide generation from arginine (*Murray, 2016*). Amino acid auxotrophy of immune cells also forms checkpoints whereby distinct amino acid metabolizing enzymes expressed in different cell types control cell proliferation, fate, and function (*Murray, 2016*). These regulated enzymes metabolize arginine (Arginases 1 and 2 and nitric oxide synthase two or iNOS) and tryptophan (indole-2,3-dioxygenases, IDO1 and IDO2). The expression of each enzyme is regulated by diverse external cues, such as cytokines and microbial signaling through toll-like receptors, which restricts their activity to specific phases in immune responses. Because each enzyme can locally deplete essential amino acids, considerable attention has been focused on the role of arginases and IDO enzymes in suppressing access of proliferating T cells to amino acids in tumor microenvironments, a process linked with T cell tolerance and spreading or 'infectious tolerance' to abate ongoing T cell-mediated immunity (*Cobbold et al., 2009*). Tolerance to malignant cells is a hallmark of cancer, and thus drugs that target arginases and IDO have been produced and tested in the context of immune enhancement of anti-tumor immunity (*Steggerda et al., 2017*; *Günther et al., 2019*).

Another immune-associated enzyme that metabolizes amino acids and linked to immune suppression in cancer is interleukin-4-induced-1 (IL4i1), which was discovered in a screen for IL4-regulated cDNAs in B cells (*Chu and Paul, 1997*). Unlike arginases and the IDO enzymes, IL4i1 is secreted (*Boulland et al., 2007*), suggesting it contributes to immune regulation in the extracellular milieu. IL4i1 is a dimeric FAD-dependent oxidoreductase with considerable similarity to L-amino acid oxidases (LAAOs) in snake venoms (*Murray, 2016*). Mammalian IL4i1 metabolizes L-Phe and other aromatic amino acids to their keto-acid forms with coincident production of $H_2O_2$ and ammonia (*Boulland et al., 2007*). However, limited information exists about the enzymology and functions of either IL4i1 or LAAOs, their substrate ranges, kinetics, and the role of downstream metabolites formed from amino acids.

Venomous snakes are divided into two broad groups: viperids (e.g., vipers, rattlesnakes, bushmasters) and elapids (cobras, taipans, mambas, sea snakes) based on their predation strategies and anatomy. Viperids have retractable fangs and powerful jaws and use large venom volumes to inactivate prey. Elapids have fixed fangs and strike to deliver small but potent venoms. A remarkable finding that emerged from mass spectrometry analysis of many different venoms concerns vast qualitative and quantitative venom protein diversity, even of closely related snake species (*Tasoulis and Isbister, 2017*; *Villar-Briones and Aird, 2018*). For example, cobra venoms contain many types of toxins, enzymes, and neurotoxins while viperid venoms are enriched in phospholipases (*Pla et al., 2017*; *Suryamohan et al., 2020*). However, all analyzed snake venoms contain a LAAO in different proportions depending on the snake, raising the question of the role of this enzyme in fast-acting venoms.

Collectively, numerous elements of IL4i1 and LAAO biology remain unresolved. These include the mechanistic basis of the association between IL4i1 and different cancers (*Carbonnelle-Puscian et al., 2009*; *Cousin et al., 2015*; *Sadik et al., 2020*), the mode of action of LAAOs in snake venoms, and how this compares to mammalian IL4i1, the cell types that express IL4i1 and relative role of IL4i1-mediated microenvironmental amino acid depletion versus product generation. To illuminate new aspects of IL4i1/LAAO biology, we initiated a comparative study between mammalian expressed recombinant forms of snake (Indian cobra, *Naja naja*) LAAO and mammalian IL4i1. This approach allowed us to establish that snake LAAO is an $H_2O_2$ 'bomb' that rapidly kills mammalian cells. By contrast, IL4i1 also generates $H_2O_2$ but is not cell-lethal. Instead, mammalian IL4i1 suppresses cell death by ferroptosis via product generation from L-Trp and L-Tyr. Most significantly, we show that indole-pyruvate (I3P) and 4-hydroxy-phenylpyruvate (4HPP) from L-Trp and L-Tyr, respectively, act as radical scavengers and activate cell protective pathways that suppress ferroptosis. We propose that components of the pro-tumor effects of IL4i1 are linked to local anti-ferroptotic effects on transformed cells.

## Results

### Venom LAAO but not mammalian IL4i1 induces cell death in an $H_2O_2$-dependent way

IL4i1 expression and secretion is generally associated with myeloid cells and B cells, the latter of which is the original cell type where IL4i1 was identified (*Chu and Paul, 1997*). As viperid and elapid LAAOs are also secreted proteins that accrue in the venom gland and bear strong sequence identity to mammalian IL4i1 (*Figure 1—figure supplement 1A*) and catalyze the same reactions (*Figure 1A*), we first asked if the cellular biochemistry of snake LAAO would help illuminate the biology of mammalian IL4i1. We expressed the authentic *N. naja* LAAO cDNA (*Suryamohan et al., 2020*) in mammalian cells (*Figure 1B,C*). Following purification and testing the glycosylation status (*Figure 1C,D*), we added different amounts of the *N. naja* LAAO to HeLa cells and observed them over time by live cell imaging using CellTox Green dye to stain dead cells. The membranes of LAAO-treated HeLa cells began to bleb after ~5 hr, and all cells were dead within 12 hr (*Figure 1E,F*; *Video 1*). To investigate the mechanism of LAAO-mediated cell death we generated a double mutant 'enzyme-dead' *N. naja* LAAO by mutating two residues predicted to be in the catalytic domain from inference for the structure of the Malayan pit viper LAAO (*Pawelek et al., 2000*; *Moustafa et al., 2006*; *Figure 1—figure supplement 1A,B*). Like the wild-type *N. naja* LAAO, the mutant version was secreted and glycosylated as expected (*Figure 1D*) but was devoid of LAAO activity using L-Phe as a

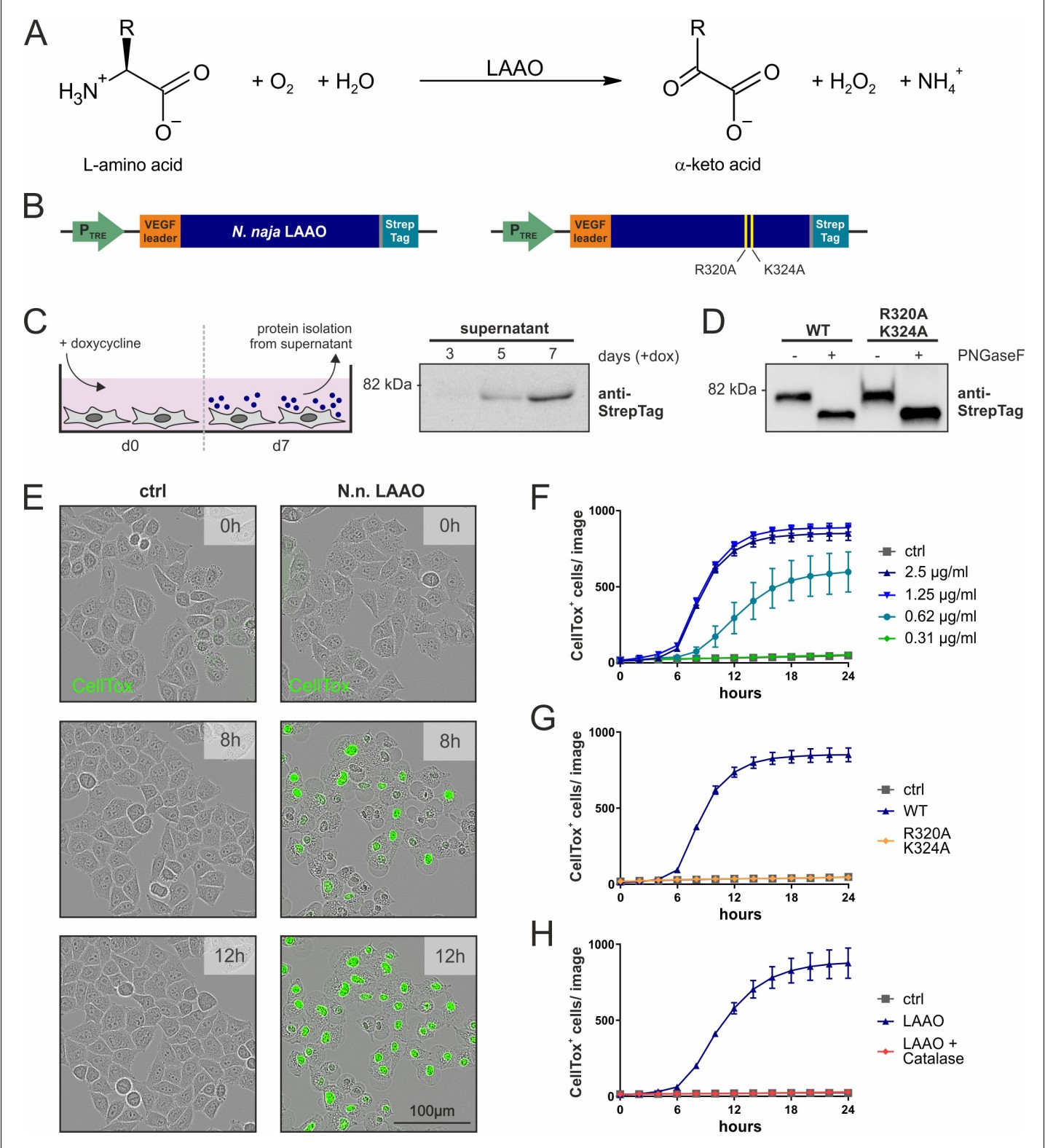

**Figure 1.** *Naja naja* LAAO is cell-lethal via $H_2O_2$. (**A**) Reaction mechanism of L-amino acid oxidases (LAAOs). (**B**) Construct design to express venom LAAOs in mammalian cells. The *N. naja* LAAO variants contain the human VEGF signal sequence and a C-terminal Strep-tag to facilitate purification. Mutations R320A and K324A ablate catalytic activity. (**C**) Purification strategy for LAAO, which is isolated from the cell supernatant. (**D**) Immunoblotting of purified recombinant proteins. LAAO or the enzyme-dead variant are glycosylated in their secreted forms. (**E**) Representative microscopy images of HeLa cells stained with the cell death dye CellTox following addition of 2.5 µg/ml *N. naja* LAAO. (**F**) Quantification of cell death across time induced by

*Figure 1 continued on next page*

*Figure 1 continued*

*N. naja* LAAO. (**G**) *N. naja* LAAO R320A and K324A enzyme-dead version fails to induce death. 2.5 µg/ml of WT and mutant enzyme was added. (**H**) Addition of catalase (25 µg/ml) blocks cell death induced by *N. naja* LAAO (2.5 µg/ml). (**F–H**): n = 3 biological replicates; the graphs are representative for three independent experiments. All error bars represent standard deviation.

The online version of this article includes the following source data and figure supplement(s) for figure 1:

**Source data 1.** Source data for the graphs in *Figure 1*.
**Figure supplement 1.** *Naja naja* LAAO is cell-lethal via $H_2O_2$.

substrate (*Figure 1—figure supplement 1C*). Following purification, the enzyme dead *N. naja* LAAO did not induce death of HeLa cells, suggesting that an enzymatic function of LAAOs is responsible for cell toxicity rather than binding to protein(s) associated with HeLa cells that subsequently conveyed a death signal (*Figure 1G*).

All LAAOs and IL4i1 are predicted to produce $H_2O_2$ as a product of their oxidoreductase activity on amino acids (*Figure 1A*). Therefore, we next asked if $H_2O_2$ was responsible for mammalian cell death. We added the wild-type *N. naja* LAAO in combination with catalase, which decomposes $H_2O_2$. The addition of catalase protected HeLa cells from *N. naja* LAAO-induced death (*Figure 1H*).

We next asked if mammalian-expressed recombinant versions of human or mouse IL4i1 had similar properties to the *N. naja* LAAO. Unlike the cobra LAAO, neither equivalent concentrations of mouse nor human IL4i1 induced cell death in HeLa cells at the tested concentrations (*Figure 2A,B*). Therefore, we used the human RS4;11, a B cell leukemia cell line, which is exquisitely sensitive to *N. naja* LAAO in the ng/ml range (*Figure 2C*). Mammalian IL4i1 required concentrations more than 30-fold higher than the *N. naja* LAAO to cause cytotoxicity (*Figure 2D*). We observed cell death induced by mIL4i1 at concentrations higher than 2 µg/ml, which was, consistently to our observations with the snake venom LAAO, ablated in an enzyme-dead K351A mIL4i1 mutant (*Figure 2—figure supplement 1*). Thus, by contrast to snake venom LAAO, it is likely that mediating cytotoxicity is not the main purpose of mammalian IL4i1. Nevertheless, we cannot exclude the possibility that IL4i1 may have cell toxicity in a circumstance where enzyme accumulation was concentrated in the extracellular milieu to µg/ml ranges.

To understand the biochemical differences between the mammalian and snake enzymes we performed a comparative kinetic and substrate utilization analysis of the *N. naja* LAAO and mammalian IL4i1. When we compared the *N. naja* LAAO with human and mouse IL4i1 using all individual 20 amino acids using an $H_2O_2$ assay, the hydrophobic and aromatic preference of the enzymes was evident (*Figure 2E*). However, *N. naja* LAAO generated much more $H_2O_2$ in the presence of its amino acid substrates within the analyzed time window of 5 min, indicating a higher enzymatic activity, and also metabolized L-Leu compared to human and mouse IL4i1. Taken together, these data indicate that snake LAAO has a superior reaction rate and substrate range compared to mammalian IL4i1 suggesting that upon envenomation, rapid $H_2O_2$ production contributes to venom efficiency. Given that human IL4i1 does not induce cell death via $H_2O_2$ (*Figure 2D*) or another mechanism in the time frame of the experimental system used, we next asked whether the keto-acid products of amino acid metabolism influenced cell state. Therefore, we investigated whether we could detect IL4i1-associated amino acid metabolites when incubating cell culture medium with the recombinant enzyme. Using an untargeted metabolomics approach, we found human IL4i1 to cause significant increases in indole-3-pyruvate (I3P), 4-hydroxy-phenylpyruvate (4HPP), phenylpyruvate (PP), 2-keto-4-methylthiobutyric acid (KMBA), and 2-oxoarginine (2OA) (from L-Trp, L-Tyr, L-Phe, L-Met, and L-Arg, respectively) (*Figure 2F*). We note that

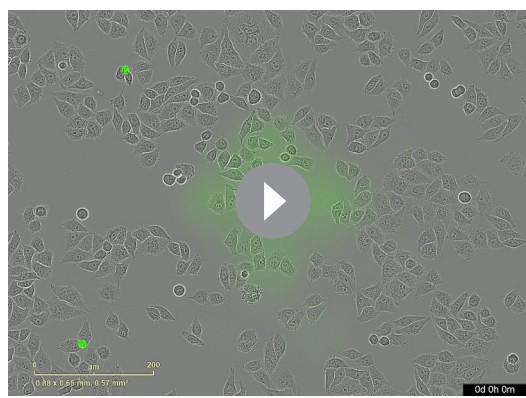

**Video 1.** Video file of *Naja naja* LAAO action on HeLa cells.
https://elifesciences.org/articles/64806#video1

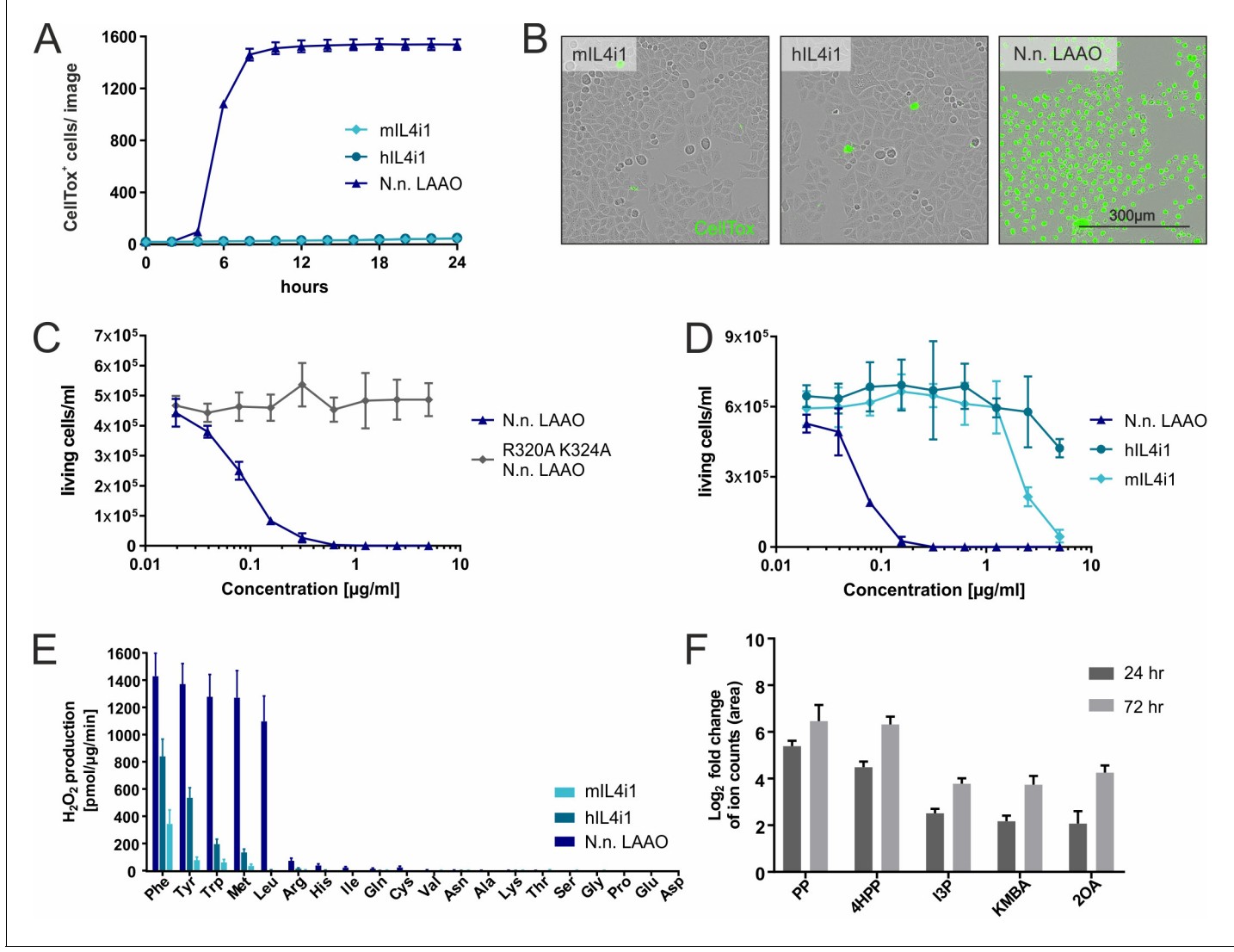

**Figure 2.** Substrate ranges and properties of IL4i1 compared to LAAO. (A) Human or mouse IL4i1 fail to induce cell death of HeLa cells at equivalent concentrations to *Naja naja* LAAO (all enzymes added at 2.5 µg/ml) (n = 3 biological replicates). (B) Representative microscopy images with CellTox staining from an experiment similar to that in (A). (C) Activity of *N. naja* LAAO or its enzyme-dead mutant to RS4;11 human leukemia cells. $1 \times 10^4$ cells were plated in round-bottom 96-well plates and treated with the indicated concentrations of LAAO enzyme. The final viable cell number was determined 72 hr after treatment (n = 3 biological replicates). (D) As in (C) comparing human and mouse IL4i1 to *N. naja* LAAO (n = 4 biological replicates). (E) Comparative enzyme activity of human and mouse IL4i1 versus *N. naja* LAAO using single amino acids. The graph includes three independent experiments. (F) Log2-fold change of ion counts compared to untreated medium of significantly increased amino acid metabolites (p<0.01) detected by untargeted metabolomics in DMEM incubated for 24 or 72 hr with 1 µg/ml of human IL4i1; n = 3 biological replicates. All error bars represent standard deviation.

The online version of this article includes the following source data and figure supplement(s) for figure 2:

**Source data 1.** Source data for the graphs in *Figure 2*.

**Figure supplement 1.** Substrate ranges and properties of IL4i1 compared to LAAO.

several of these metabolites including I3P can be generated by microbiota or in infection and these could be present in serum (*Dodd et al., 2017*). However, they were significantly increased in medium incubated with IL4i1, suggesting the enzyme is actively generating these metabolites. These data are indicative of the overall biology of immune-associated amino acid metabolizing enzymes that can regulate neighboring cells via essential acid depletion and product generation.

## IL4i1 generates metabolites that activate stress protection pathways

To investigate whether the products of the IL4i1 reaction were involved in modulating cell state, we performed RNAseq analysis on human THP-1 monocytic cells treated with I3P, 4HPP, or PP for 24 hr (*Figure 3A–C*). THP-1 cells were used in these experiments because IL4i1 has highest expression in myeloid cells (*Boulland et al., 2007*; *Rieckmann et al., 2017*). Thus, we reasoned that autocrine–paracrine effects of IL4i1-mediated amino acid metabolism would be detected in this cell system. We found that PP had an almost undetectable effect on the global quality and quantity of THP-1 mRNAs. By contrast, I3P, and to a lower extent 4HPP, induced the expression of a suite of mRNAs including those encoding SLC7A11, NQO1, ATF4, CYP1B1, and different members of the AKR1C family (*Figure 3*). Most of these mRNAs encode proteins involved in cellular stress, especially in oxidative stress response (*Figure 3D*), and are regulated by the ATF4, Nrf2, and AhR pathways (*Zgheib et al., 2018*; *Liu et al., 2020*).

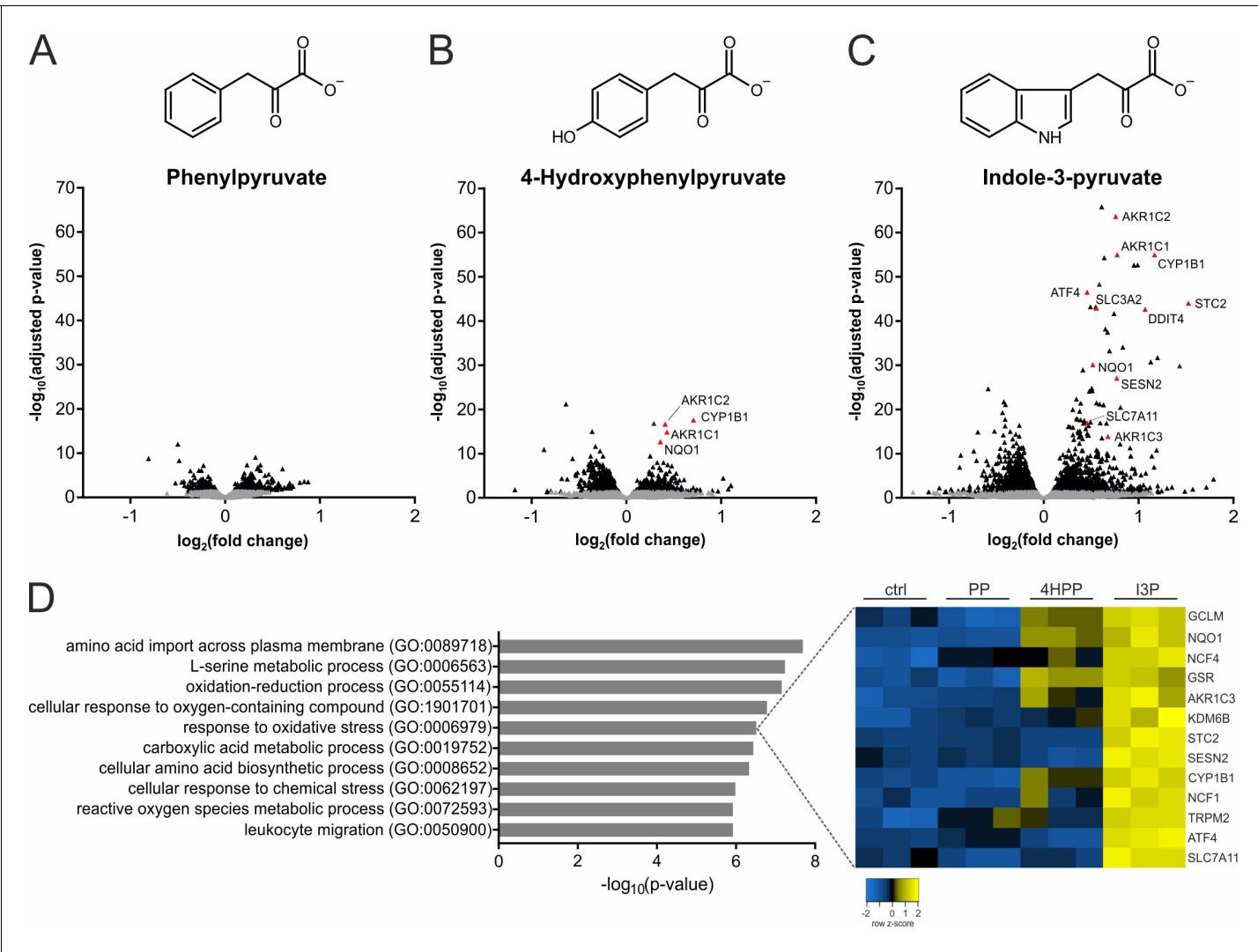

**Figure 3.** IL4i1 products induce specific stress and redox-protective gene expression programs. (A–C) THP-1 human monocytic cells were trated for 24 hr with PP, 4HPP, or I3P (200 μM) and RNAseq analysis was used to quantify gene expression changes compared to untreated controls. (D) Top hits of GO term overrepresentation analysis of the most significantly (p-value cutoff $p<10^{-9}$) upregulated genes by I3P treatment (left). The full table of significant GO terms is provided (*Supplementary file 1*). Heat map of genes detected in the analysis under the term GO:0006979 (right).

## IL4i1 metabolites suppress ferroptosis

Because I3P and 4HPP induced a selective gene expression profile consistent with stress protection and particularly defense against oxidative stress, we reasoned these IL4i1 metabolites could intersect at some point in the pathways that control the suppression of ferroptosis, a programmed oxidative cell death characterized by massive lipid peroxidation (*Stockwell et al., 2017*; *Conrad and Pratt, 2019*; *Tang and Kroemer, 2020*; *Figure 4A*). We therefore performed live cell imaging to monitor ferroptosis in HeLa cells induced by erastin (targeting SLC7A11, the cysteine–glutamate exchanger) and RSL3, which inhibits GPX4, required for glutathione-mediated lipid peroxide decomposition. Ferrostatin-1 (Fer-1) was used as control for ferroptosis protection (*Dixon et al., 2012*). We pre-treated or concurrently treated the cells with I3P, 4HPP, or PP, with the expectation that PP could serve as a type of negative control consistent with its limited effect on gene expression in THP-1 cells. Indeed, while PP had no protective effect, we found I3P suppressed ferroptosis in a titratable way (*Figure 4B,C*; *Figure 4—figure supplement 1*), which was accompanied by significantly decreased lipid peroxidation measured with C11-BODIPY (*Figure 4D*). 4HPP exhibited weaker anti-ferroptotic properties than I3P and required 24 hr pre-incubation to potently protect from RSL3-mediated cell death (*Figure 4B,C*; *Figure 4—figure supplement 1D*). In contrast to I3P, L-Trp failed to protect cells from ferroptosis, indicating that the conversion of L-Trp to I3P by IL4i1, is required for the protection (*Figure 4—figure supplement 1E*). Importantly, I3P was active in a concentration range starting from ~50 µM (*Figure 4—figure supplements 1* and *2*). Considering blood concentrations of L-Trp and L-Tyr (~70 and ~90 µM, respectively) (*Geisler et al., 2015*), I3P was detected at ~80 µM (*Dodd et al., 2017*), and that IL4i1 is a secreted protein and active in the extracellular milieu, the protective concentrations of I3P and 4HPP are within a range to operate on neighboring cells, especially in limited perfusion cellular environments such as lymph nodes where numerous IL4i1-positive cells reside. We verified our observations made in HeLa cells in HT1080 cells, a human fibrosarcoma cell line frequently used in ferroptosis studies (*Figure 4—figure supplement 2A–C*). Additionally, measuring cell death using 7-AAD and Annexin-V staining via flow cytometry, we confirmed the anti-ferroptotic effect of I3P toward erastin, RSL3, an additional ferroptosis inducer termed FINO2, and the thioredoxin-reductase inhibitor ferroptocide in murine NIH3T3 (*Figure 4E*) and HT1080 cells (*Figure 4—figure supplement 2D*), showing that I3P acts as a general, species-independent protector from ferroptosis.

## IL4i1 metabolite I3P protects from ferroptosis by radical scavenging and activation of anti-oxidative stress pathways

We next investigated the mechanisms associated with IL4i1 metabolite I3P-mediated ferroptosis suppression. As described above (*Figure 3*), we found that I3P induced a network of genes involved in the protection from oxidative stress in THP-1 cells (*Figure 3*). However, there is also evidence for I3P acting as a free radical scavenger (*Politi et al., 1996*). Therefore, we tested the radical scavenging activity of the IL4i1 amino acid metabolites PP, 4HPP, and I3P toward the stable radical diphenyl-2-picrylhydrazyl (DPPH). Ascorbic acid and Fer-1, which have been shown to scavenge the DPPH radical (*Niki, 1991*; *Dixon et al., 2012*) were used as positive controls at 200 µM. By contrast to PP, we found both, 4HPP and I3P to exhibit radical scavenging properties, with I3P having the highest scavenging potency of the tested IL4i1 metabolites (*Figure 5A*). Thus, the anti-ferroptotic activity could either depend on radical scavenging or the activation of anti-oxidative stress pathways. To unravel these two potential mechanisms of protection we focused on I3P, the most potent anti-ferroptotic metabolite. We hypothesized that a scavenging effect would require the direct presence of I3P, whereas after the induction of a protective gene network I3P would not have to be present anymore at the time point of ferroptosis induction, a concept validated for other ferroptosis inhibitors (*Tang and Tang, 2019*). Therefore, we took advantage of the fluorescent properties of I3P to monitor its presence in the cells using flow cytometry (*Figure 5—figure supplement 1*). After preincubating HeLa cells for 24 hr with I3P, an increase in fluorescence measured in the AmCyan channel was observed as compared to cells incubated in control medium. We washed and removed the I3P-containing medium and tracked the decline in intracellular I3P fluorescence over time (*Figure 5B*). Four hours after I3P removal, the cells reached fluorescence levels similar to control cells, indicating the deprivation of intracellular I3P. Thus, we asked whether I3P was still protective under this condition (*Figure 5C*). Strikingly, we found I3P to retain its protective effect toward

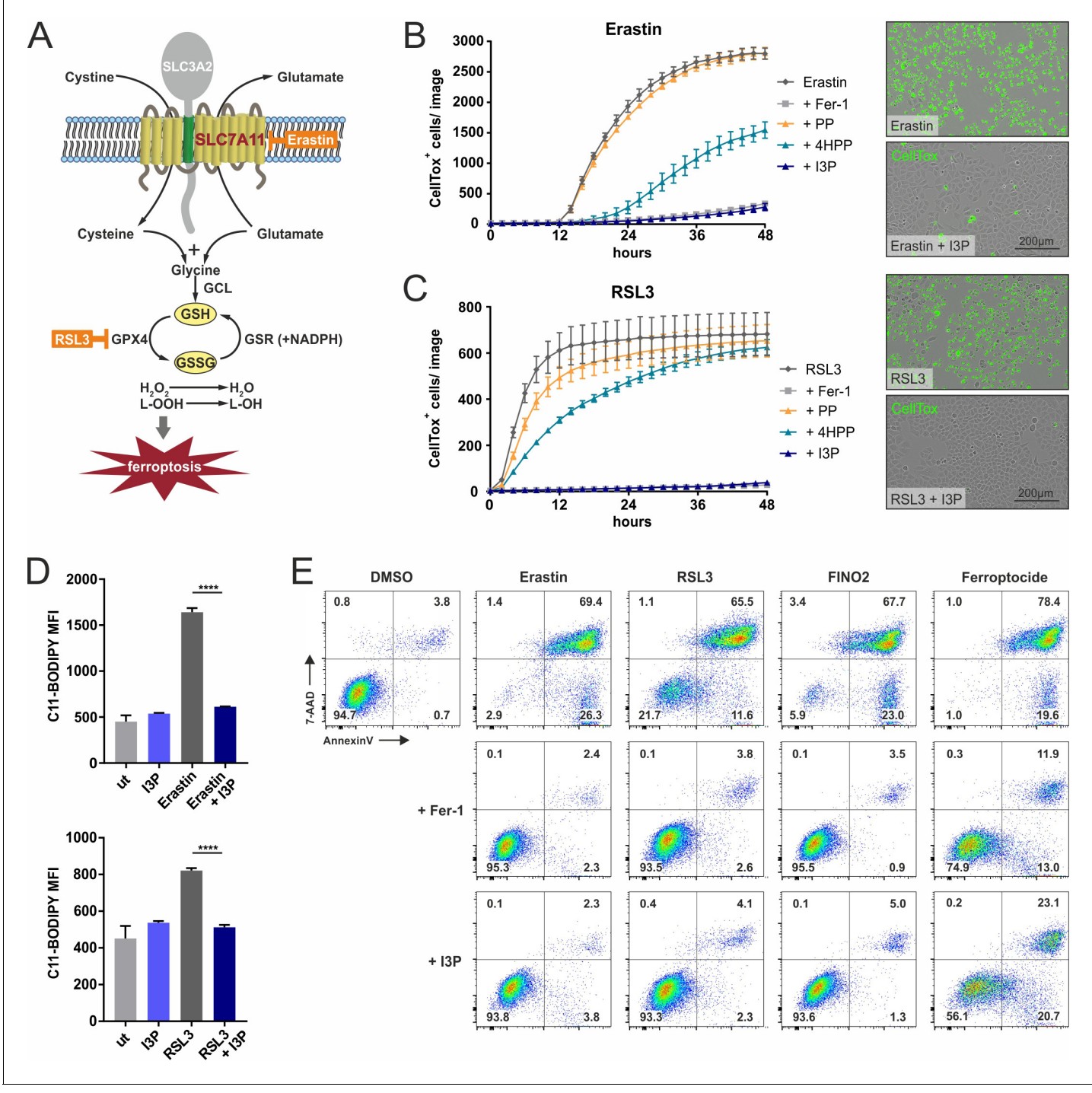

**Figure 4.** IL4i1 metabolites form an anti-ferroptotic hierarchy. (**A**) Simplified schema of ferroptosis control showing the points of chemical perturbation by erastin and RSL3. (**B**) Quantification of cell death of HeLa cells treated with the ferroposis inducer erastin in the presence of 200 μM PP, 4HPP, or I3P by live cell imaging using CellTox straining. Ferrostatin-1 (Fer-1) was added as a control to block erastin-induced death and has an equivalent suppressing effect as I3P. Right panel, representative CellTox staining images from an experiment similar to (**B**). (**C**) As in B, using RSL3 to induce ferroptosis. B,C: n = 3 biological replicates. The graphs are representative for three independent experiments. (**D**) I3P blocks lipid peroxidation induced by erastin and RSL3 determined by flow cytometry using C11-BODIPY. n = 3 biological replicates; data were analyzed by one-way ANOVA with Tukey's multiple comparisons test; ****p<0.0001. All error bars represent standard deviation. (**E**) Flow cytometry analysis of the anti-ferroptotic activity of I3P (200 μM) using Fer-1 as positive control. Murine NIH3T3 cells were treated with erastin, RSL3, FINO2, or ferroptocide to induce ferroptosis in the absence or presence of Fer-1or I3P and death quantified by 7-AAD and Annexin-V staining. See 'Materials and methods' for details of reagent concentrations and timing. The plots are representative for two independent experiments.

*Figure 4 continued on next page*

*Figure 4 continued*

The online version of this article includes the following source data and figure supplement(s) for figure 4:

**Source data 1.** Source data for the graphs in *Figure 4*.
**Figure supplement 1.** IL4i1 metabolites form an anti-ferroptotic hierarchy.
**Figure supplement 2.** IL4i1 metabolites form an anti-ferroptotic hierarchy.

erastin-induced ferroptosis, but a complete loss of protection from RSL3-mediated cell death (*Figure 5D,E*). These results suggest that I3P uses at least two different modes to protect cells from ferroptosis: (i) free radical scavenging and (ii) the induction of specific protective gene expression. Protection from RSL3 mainly depends on radical scavenging, whereas the persistent protection from erastin after I3P removal, indicates the major involvement of anti-oxidative gene expression under these conditions.

Ferroptosis can be initiated at different levels of cellular metabolism including perturbations in cysteine import needed for glutathione biosynthesis, glutathione homeostasis, which requires cysteine, glycine, glutamate, and NADPH, iron-heme homeostasis, and the relative balance between ROS generation and decomposition (*Stockwell et al., 2017*; *Conrad and Pratt, 2019*; *Tang and Kroemer, 2020*). In contrast to RSL3, which directly inhibits the detoxification of lipid peroxides by GPX4, erastin disturbs the cellular redox balance by inhibiting cysteine import (*Figure 4A*). Since this balance can be restored by the induction of anti-oxidative gene networks, we tested whether I3P modulates the intracellular ratio of reduced and oxidized GSH (GSH/GSSG ratio). Indeed, we observed a significant increase in the GSH/GSSG ratio in HeLa cells upon I3P treatment (*Figure 5F*). Additionally, we found an I3P-dependent increase of SLC7A11 protein under steady state conditions in HeLa cells treated for 24 hr (*Figure 5G*) coinciding with the transcriptional increase of *SLC7A11* in THP-1 cells (*Figure 3*). Erastin itself upregulates *SLC7A11* (*Dixon et al., 2014*) as a compensatory mechanism to elevate intracellular cysteine. In the presence of I3P, however, SLC7A11 upregulation was strongly enhanced (*Figure 5G*) and also HO-1, a target gene of anti-oxidative Nrf2 signaling, was also upregulated, suggesting that I3P increases anti-oxidative gene expression under erastin-mediated redox stress. In line, we also found 4HPP upregulated SLC7A11 and HO-1 in the presence of erastin (*Figure 5—figure supplement 2A*). HO-1 was reported to have both detrimental and protective functions in ferroptosis (*Kwon et al., 2015*; *Chiang et al., 2018*). Therefore, we tested whether HO-1 activity was required for the anti-ferroptotic effect of I3P upon erastin treatment. We observed that HO-1 inhibitors, ZnPP and ketoconazole (*Yang et al., 2018*), reduced the protective effect of I3P (*Figure 5—figure supplement 2B,C*). However, genetic perturbation via siRNA-mediated knockdown of HO-1 did not interfere with I3P-mediated ferroptosis protection (*Figure 5—figure supplement 2D,E*), suggesting that the inhibitors are not completely specific and may target other proteins such as HO-2 or other heme-dependent enzymes. Taken together, our results indicate that, apart from ROS scavenging, the IL4i1 metabolite I3P (and to some extent also 4HPP) protects from ferroptosis by triggering a complex network of intracellular anti-oxidant signaling pathways resulting in increased GSH levels.

## IL4i1 generates a ferroptosis-suppressive milieu

A key implication from or finding that I3P and 4HPP are anti-ferroptotic IL4i1 metabolites, concerns the effects of IL4i1 in generating an anti-ferroptotic environment that would promote redox stress survival. To test this concept, we added 1 µg/ml of recombinant WT or K351A mutant, enzymatically inactive (*Figure 2—figure supplement 1B*), murine IL4i1 to DMEM and incubated for 72 hr at 37°C allowing the enzyme to produce its metabolites. We compared the gene expression signature of HeLa cells left untreated, incubated for 24 hr with IL4i1 conditioned medium (retaining the enzyme) or 200 µM I3P. Hierarchical clustering of significantly regulated genes showed concordance of differentially expressed genes in cells treated with I3P or medium containing the WT IL4i1 enzyme, whereas the gene expression in HeLa cells treated with medium containing the enzyme-dead mutant IL4i1 clustered with the untreated control (*Figure 6A*). Thus, the enzymatic activity IL4i1 is required for the changes in the gene expression and the generation of I3P seems to be one major factor for the IL4i1-mediated gene expression profile, which is likely enhanced by the additional generation of 4HPP and low levels of $H_2O_2$. Indeed, more than 90% of the most significantly upregulated genes

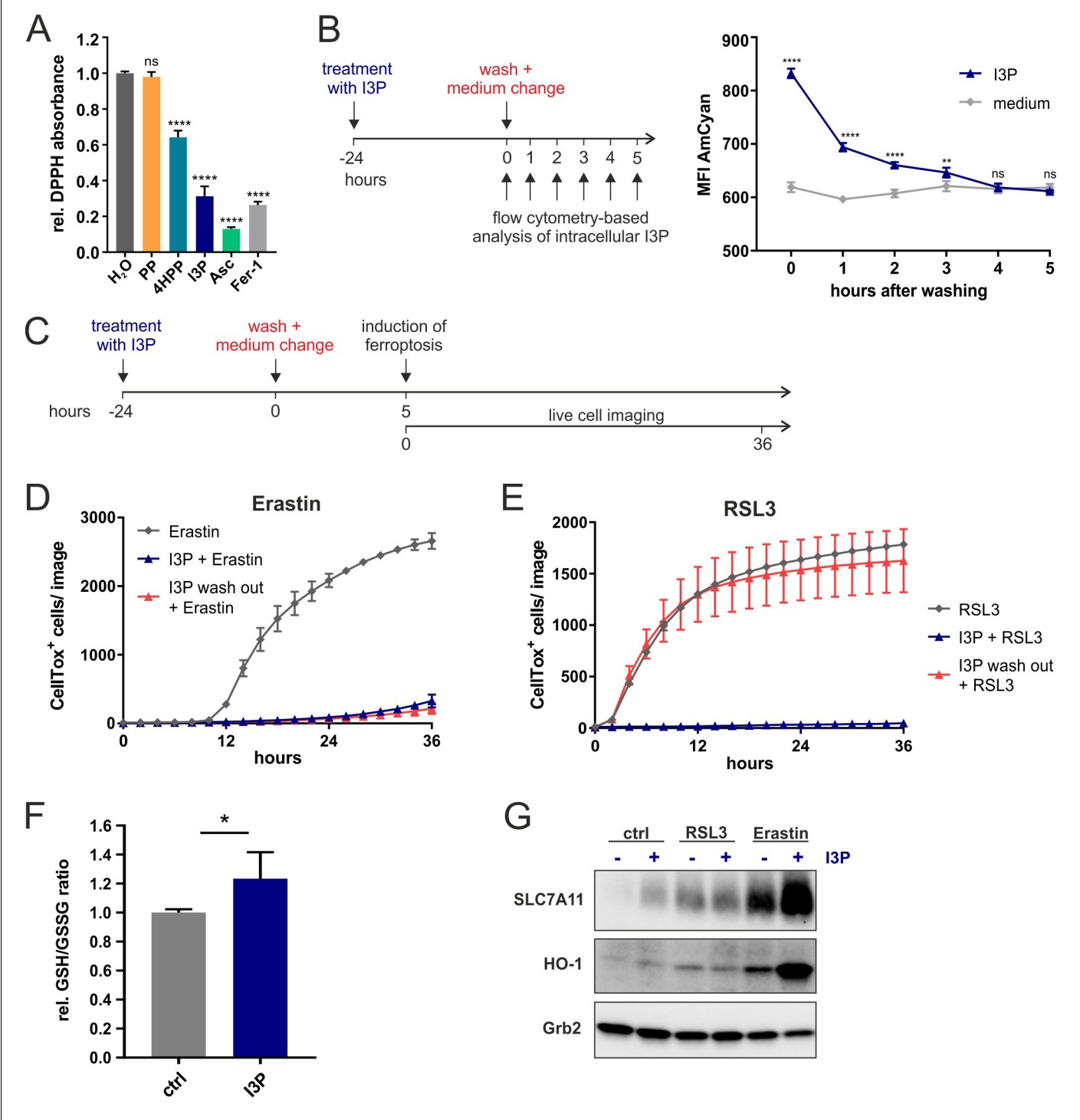

**Figure 5.** Anti-ferroptotic mechanisms of I3P. (**A**) Cell-free scavenging activity of 200 µM PP, 4HPP, I3P, ascorbic acid (Asc), and Fer-1 determined by changes in the absorbance at 517 nm of the stable radical DPPH relative to $H_2O$ control. n = 4 Technical replicates; the graphs are representative for three independent experiments. Data were analyzed by one-way ANOVA with Dunnett's multiple comparisons test; ****p<0.0001; ns = not significant. (**B**) Design of a wash-out experiment to quantify intracellular I3P. HeLa cells were incubated for 24 hr with or without 200 µM I3P. After washing-out the I3P-containing medium, remaining intracellular I3P was determined by flow cytometry at the indicated time points. n = 3 biological replicates; the graphs are representative for three independent experiments. Data were analyzed by two-way ANOVA with Sidak's multiple comparisons test; **p<0.01; ****p<0.0001; ns = not significant. (**C**) Adaptation of the assay in (**B**) to quantify direct versus indirect protective effects of I3P. (**D**) I3P retains protective activity against erastin-induced ferroptosis after wash-out. (**E**) I3P only protects against RSL3-induced ferroptosis if present in cells during the

*Figure 5 continued on next page*

*Figure 5 continued*

time of GPX4 inhibition. D,E: n = 2 biological replicates; the graphs are representative for three independent experiments. (**F**) GSH/GSSG ratio in HeLa cells treated with 200 µM I3P for 24 hr relative to untreated control. n = 4 biological replicates; data were analyzed using an unpaired *t*-test; *p<0,05. All error bars represent standard deviation. (**G**) I3P induces the expression of SLC7A11 and HO-1 at protein level. HeLa cells were treated with I3P in the absence or presence of erastin or RSL3 for 24 hr and SLC7A11 and HO-1 levels determined by immunoblotting. Grb2 was used as loading control. The online version of this article includes the following source data and figure supplement(s) for figure 5:

**Source data 1.** Source data ffor the graphs in *Figure 5*.
**Figure supplement 1.** Anti-ferroptotic mechanisms of I3P.
**Figure supplement 2.** Anti-ferroptotic mechanisms of I3P.

(adjusted p-value<$10^{-9}$) upon I3P treatment overlapped with the most significantly upregulated genes by WT IL4i1 (*Figure 6B*) – many of them representing target genes of Nrf2 and AhR signaling, and involved in oxidative stress protection.

Moreover, monitoring I3P fluorescence via flow cytometry, we detected a dose dependent increase in the AmCyan fluorescence in cells treated with WT IL4i1-conditioned medium, but not in cells treated with the enzyme-dead mutant, confirming that IL4i1-generated I3P was taken up by the cells (*Figure 6C*). In line with our previous findings, we observed this to be linked to a dose-dependent increase in HO-1 and SLC7A11 (*Figure 6D*), which was absent when using the control medium or the K351A mutant enzyme (*Figure 6E*). To examine whether cells treated with the IL4i1-conditioned medium are protected from ferroptosis, we treated cells with erastin or RSL3 to induce ferroptosis after transferring the medium to the cells. Importantly, whereas the mutant enzyme had no protective effect, WT IL4i1 inhibited ferroptosis in a titratable manner (*Figure 6F*, *Figure 6—figure supplement 1A*), indicating that IL4i1 can create a ferroptosis-suppressive milieu due to its enzymatic activity. To additionally confirm that these effects do not require IL4i1 to bind to any receptor, but the changes in the metabolic environment are sufficient, we used filtration to remove IL4i1 from the conditioned medium before addition to the HeLa cells, which also resulted in upregulation of HO-1 and SLC7A11 and protection from ferroptosis (*Figure 6—figure supplement 1B–D*). Since amino acid concentrations in cell culture medium do not reflect the physiological levels, we verified that the protective effect of IL4i1 was also retained in medium containing human plasma concentrations of amino acids and glucose as described by *Leney-Greene et al., 2020*; *Figure 6—figure supplement 1E*. These data indicate that after IL4i1 is secreted, it generates I3P, 4HPP, and other metabolites that have an autocrine–paracrine anti-ferroptotic effect. Thus, whereas snake venom LAAO quickly generates cytotoxic levels of $H_2O_2$, IL4i1 produces non-lethal amounts of $H_2O_2$ and amino acid metabolites protecting cells from oxidative ferroptotic cell death (*Figure 6G*). Therefore, we tested if IL4i1 could also protect cells from LAAO toxicity. As in *Figure 6F*, we transferred medium pre-incubated for 72 hr with IL4i1 to HeLa cells and added a toxic concentration (1.25 µg/ml; *Figure 1F*) of snake venom LAAO. However, IL4i1 could not prevent cells from LAAO toxicity (*Figure 6—figure supplement 1F*). In concordance with our previous observations regarding ferroptosis protection we confirmed that the venom LAAO does not induce this mode of cell death (*Figure 6—figure supplement 1G*). Thus, we speculate that IL4i1 can rather protect from endogenous ROS accumulation, but not from the rapid exogenous $H_2O_2$ accumulation induced by the venom protein.

## Discussion

Our results indicate that snake venom LAAOs and mammalian IL4i1 diverged in their functions to kill or protect cells, respectively. We found that a venom LAAO killed mammalian cells via $H_2O_2$ generation. *N. naja* LAAO oxidized five amino acids to generate $H_2O_2$, which was lethal to mammalian cells. *N. naja* LAAO was far more potent than its mammalian counterparts, even though the substrate range was almost similar to IL4i1 (with the main exception of L-Leu being metabolized by LAAO). These data are consistent with the notion that venom LAAOs evolved to rapidly metabolize amino acids from their target, and work in concert with other cytotoxic enzymes, such as phospholipases, to increase cell permeability and death in and around the site of envenomation. We note that although LAAOs are almost universal in viperid and elapid venoms, their relative amounts differ remarkably across snake species, consistent with the prey- dependent evolutionary pressure to tailor

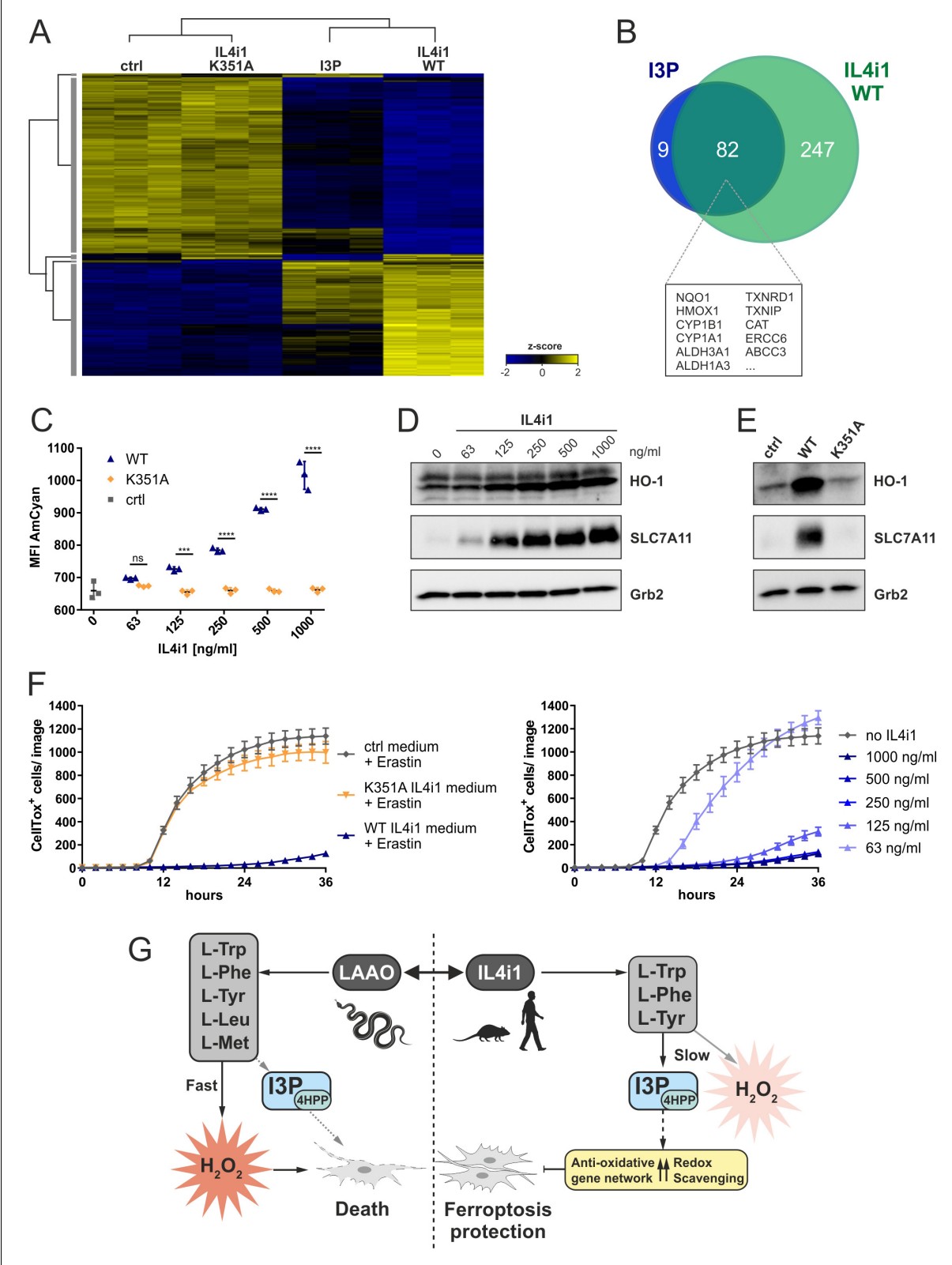

**Figure 6.** IL4i1 generates an anti-ferroptotic milieu. (**A**) Heat map of differentially expressed genes in HeLa cells after 24 hr incubation with IL4i1 (WT or K351A inactive mutant [*Figure 2—figure supplement 1*])-conditioned DMEM, 200 μM I3P, or untreated control medium. (**B**) Overlap of most significantly upregulated genes as compared to the untreated HeLa cells (adjusted p-value<10$^{-9}$) in HeLa cells treated with I3P and WT IL4i1 conditioned medium. Many of the overlapping genes are associated with Nrf2 and AhR signaling. (**C**) I3P uptake by HeLa cells was quantified after 24

*Figure 6 continued on next page*

Figure 6 continued

hr of incubation with IL4i1 conditioned DMEM by flow cytometry. n = 3 biological replicates; the graph is representative for three independent experiments. Data were analyzed by two-way ANOVA with Sidak's multiple comparisons test; ***p<0.001; ****p<0.0001; ns = not significant. (D, E) HO-1 or SLC7A11 expression was determined by immunoblotting following transfer of complete DMEM media treated with increasing concentrations of IL4i1 or enzyme-dead K351A mutant for 24 hr. (F) Quantification of erastin-induced ferroptosis in HeLa cells in the presence of IL4i1 conditioned DMEM. WT but not K351A mutant IL4i1 conditioned medium (1 µg/ml) suppressed ferroptosis (left). The anti-ferroptotic effect of IL4i1 is concentration dependent (right). n = 2 biological replicates; the graphs are representative for three independent experiments. All error bars represent standard deviation. (G) Schematic representation of the death-inducing versus death-protection mechanisms of venom LAAO versus mammalian IL4i1.

The online version of this article includes the following source data and figure supplement(s) for figure 6:

**Source data 1.** Source data for the graphs in *Figure 6*.
**Figure supplement 1.** IL4i1 generates an anti-ferroptotic milieu.

---

venom for the target food source (*Tasoulis and Isbister, 2017*). For example, spitting cobras use directed venom ejection as a defensive strategy (*Kazandjian et al., 2021*); whether these snakes retain the same quantities and activities of LAAOs compared to their close cousins, such as *N. naja*, may provide additional insights into venom evolution.

By contrast to venom LAAOs, we discovered that mammalian IL4i1 protected against death. We found the death protection mechanism of IL4i1 is via suppression of ferroptosis. IL4i1 mediated its anti-ferroptotic activities by overlapping and distinct mechanisms that we speculate synergize in vivo. First, we found that human and mouse IL4i1 are non-cytolytic compared to snake LAAO. Second, the products of the IL4i1 reaction, PP, 4HPP, and I3P induced the hierarchical activation of a gene transcriptional signature linked to networks of cell protective genes, including SLC7A11, CYP1B1, and AKR1C1. PP had almost no detectable effect on expression of the cell protection gene network, while I3P induced a readily identifiable cluster of mRNAs, and 4HPP was intermediate between PP and I3P. Third, IL4i1 is a secreted enzyme and therefore we expect that all the products of the reaction will be made at the same time (though their clearance and downstream metabolism may differ substantially) and could therefore work collectively to activate anti-ferroptotic effects in neighboring cells. Indeed, our media filtration experiments using media treated with IL4i1 and experiments using an enzyme-dead control, argue that the products of the IL4i1 reaction are necessary and sufficient to block ferroptosis in neighboring cells. These experiments additionally argue against a model where IL4i1 binds to other proteins and acts like a cytokine or hormone in this context (*Aubatin et al., 2018*).

Using information from the cell protective gene network induced by IL4i1 metabolites, we speculated that I3P and 4HPP but not PP, may modulate the response to redox stress. We focused on ferroptosis as this pathway is amenable to both activation and suppression by well-defined on-target drugs. We found I3P was highly protective against ferroptotic death induced by both GPX4 (RSL3) or SLC7A11 (erastin) inhibition. 4HPP had an intermediate effect on suppressing ferroptosis, while PP had no modulatory effect, coinciding closely with our RNAseq data showing the distinct magnitude of gene expression by I3P relative to either 4HPP or PP. Ferroptosis can be interrupted by distinct mechanisms leading from modulating cysteine metabolism to form glutathione (*Stockwell et al., 2017*; *Conrad and Pratt, 2019*), to detoxification of lipid peroxides. In dissecting each key step, we found I3P blocked ferroptosis by at least two distinct mechanisms. First, using a radical scavenging assay and a strategy to load ferroptosis-susceptible cells with I3P and then trace its decay, we observed that I3P has direct free radical scavenging properties. We noted that I3P has previously been shown to be a free radical scavenger in an unrelated system (*Politi et al., 1996*). Second, and independent of the radical scavenging properties, I3P induced a cell redox protective gene network, sufficient to prevent erastin-mediated ferroptosis and associated with an increased ratio of reduced GSH. Moreover, especially under erastin-mediated stress conditions, I3P elevated the activation of compensatory gene expression as indicated by increased protein amounts of SLC7A11 and HO-1, two important target genes of anti-oxidative stress pathways. Nevertheless, further studies are required to identify the main conductors of ferroptosis suppression and the upstream transcription pathways. Our RNAseq data from I3P- and IL4i1-treated cells suggest a possible involvement of ATF4 and especially Nrf2 and AhR signaling. Indeed, also a recent study suggested that the AhR was activated by I3P (*Sadik et al., 2020*). Determining the relative contributions of these pathways, however, will require a systematic evaluation in cell systems devoid of each or

even combinations of these transcription factors, since there is evidence for the overlap of the pathways (*Zgheib et al., 2018*).

So far, very little is known about IL4i1 protein levels in tissues or serum, mainly due to limitations in validated antibodies. Since IL4i1 expression is highly inducible in immune cells (*Molinier-Frenkel et al., 2019*), it is possible that inflammatory conditions are required for IL4i1 accumulation. However, a recent study reported serum IL4i1 amounts of up to 300 pg/ml in a mouse leukemia model (*Sadik et al., 2020*). Given that IL4i1 expression is often associated with tumor-associated myeloid cells (*Ramspott et al., 2018*), we speculate that especially in low-perfused tissue environments, such as found in solid tumors, local concentrations of IL4i1 can reach much higher levels, attaining a ng/ml range in which we found IL4i1 to mediate its protective effects. We also note that I3P has previously been detected in plasma at ~80 µM and was speculated to be derived from microbiome tryptophan metabolism (*Dodd et al., 2017*) and that related microbiota-derived tryptophan metabolites are associated with radio-protection (*Guo et al., 2020*). Notably, we were able to demonstrate anti-ferroptotic activity of I3P within this concentration range. Given that the microbiome and IL4i1 activate anti-ferroptosis pathways via I3P and potentially other indoles, one plausible way to account for the pro-tumor effects of IL4i1 and the pathway we discovered herein, is that local IL4i1 production in the tumor microenvironment increases the available I3P (and 4HPP) concentrations, protecting tumor cells from ferroptosis. While no direct evidence for this link exists, yet the association between IL4i1 and cancer is substantial (*Ramspott et al., 2018*; *Molinier-Frenkel et al., 2019*; *Sadik et al., 2020*). Thus, we speculate that an inhibitor of IL4i1 may be a useful anti-cancer drug, as it may sensitize cancer cells to ferroptotic death by blocking IL4i1-mediated amino acid metabolism.

# Materials and methods

## Key resources table

| Reagent type (species) or resource | Designation | Source or reference | Identifiers | Additional information |
|---|---|---|---|---|
| Chemical compound, drug | Doxycycline | Sigma | D9891 | 1 µg/ml |
| Chemical compound, drug | PNGaseF | New England Biolabs | P0704 | |
| Chemical compound, drug | Erastin | Selleckchem | S7242 | |
| Chemical compound, drug | (1S, 3R)-RSL3 | Selleckchem | S8155 | |
| Chemical compound, drug | Catalase | Sigma | C1345 | |
| Chemical compound, drug | Zinc (II) protoporphyrin IX | Sigma | 691550 | |
| Chemical compound, drug | Ketoconazole | Acros Organics | 455470010 | |
| Chemical compound, drug | Ferrostatin-1 | Sigma | SML0583 | 2 µM |
| Chemical compound, drug | Liproxstatin-1 | Selleckchem | S7699 | 1 µM |
| Chemical compound, drug | Indole-3-pyruvic acid | Sigma | I7017 | |
| Chemical compound, drug | Sodium phenylpyruvate | Alfa Aesar | H56767.06 | |
| Chemical compound, drug | 4-Hydroxy phenylpyruvic acid | Sigma | 114286 | 200 µM |

*Continued on next page*

*Continued*

| Reagent type (species) or resource | Designation | Source or reference | Identifiers | Additional information |
|---|---|---|---|---|
| Chemical compound, drug | 2,2-Diphenyl-1-pikryl-hydrazyl | Sigma | D9132 | |
| Chemical compound, drug | Horse radish peroxidase | Sigma | P8375 | |
| Peptide recombinant protein | Recombinant human IL4i1 | R and D systems | 5684-AO-020 | |
| Sequence-based reagent | HO-1 siRNA | Invitrogen | 4390824 (S6673) | 50 nM |
| Sequence-based reagent | Scrambled siRNA | Invitrogen | 4390843 | 50 nM |
| Antibody | Anti-Slc7a11 (rabbit monoclonal) | Cell Signaling Technology | 12691 | 1:1000 |
| Antibody | Anti-HO-1 (rabbit polyclonal) | Enzo | ADI-SPA-895 | 1:1000 |
| Commercial assay, kit | CellTox Green | Promega | G8731 | 1:4000 |
| Commercial assay, kit | Amplex UltraRed | Thermo Fisher Scientific | A36006 | |

## Cell lines and media

Cell lines were purchased from the American Type Culture Collection (ATCC). All cell lines were tested to be free from Mycoplasma contamination by routine PCR screening.

THP-1 and RS4;11 cells were grown in RPMI Medium 1640 containing GlutaMAX (Gibco/Life Technologies Corp., Grand Island, NY) plus 10% fetal bovine serum (FBS), 1% Penicillin-Streptomycin (Sigma, St. Louis, MO), and 0.1% 2-Mercaptoethanol (Sigma Aldrich). HeLa cells, HEK293T cells, and HT-1080 and NIH3T3 cells were grown in DMEM high glucose, pyruvate (Gibco/Life Technologies Corp., Grand Island, NY) plus 10% fetal bovine serum (FBS) and 1% Penicillin-Streptomycin (Sigma, St. Louis, MO). Medium containing human plasma concentrations of amino acids was prepared using D9800-13 Dulbecco's MEM (DMEM) Low Glucose, w/o Amino Acids, Pyruvic Acid (Powder) (USBiological) adding amino acids as previously described (*Leney-Greene et al., 2020*). All cell lines were grown in humidified tissue culture incubators at 37°C with 5% $CO_2$.

## siRNA knockdown

HeLa cells were seeded at $2 \times 10^5$ cells/well in six well plates. After 24 hr incubation at 37°C, the cells were transfected with 50 nM of HO-1 siRNA (4390824, s6673, Invitrogen) or scrambled siRNA (4390843, Invitrogen) using Lipofectamine RNAiMAX Transfection Reagent (Invitrogen) according to the manufacturer's instructions. Cells were re-plated 10 hr after transfection for the ferroptosis assay and western blot analysis.

## Recombinant protein expression and purification

HEK293T (ATCC) cell lines stably expressing Twin-Strep-tagged *N. naja* LAAO (WT and R320A, K326A double mutant) and murine IL4i1 (WT and K351A mutant) were generated using the PiggyBac transposon system (*Yusa et al., 2011*; *Li et al., 2013*). For both proteins the human VEGF leader sequence (MNFLLSWVHWSLALLLYLHHAKWSQA) was used to ensure efficient secretion. Cells were grown at a density of $1 \times 10^6$ cells per ml in FreeStyle 293 Expression Medium (Gibco, Thermo Fisher Scientific) and protein expression was induced using 1 µg/ml doxycycline. For purification of the secreted proteins the supernatant was collected 7 days after doxycycline induction and filtered using a Nalgene Rapid-Flow 75 mm bottle top filter (Thermo Scientific).

For *N. naja* LAAO purification, the supernatant was concentrated from a starting volume of 2 l to a final volume of 50 ml in 20 mM MES (pH 6.0) and 50 mM NaCl by diafiltration using a Satocon Slice 200 (Sartorius). The protein was loaded onto a Toyopearl Sulfate-650F 5 ml column (Tosoh Bioscience) for ion exchange chromatography using a linear gradient up to 20 mM MES (pH 6.0) and 2000 mM NaCl for elution. The LAAO containing fractions were further purified by incubation with Strep-TactinXT Superflow Beads for 1 hr on a rotating wheel. After washing with PBS the protein was eluted with PBS +50 mM Biotin.

To purify murine IL4i1 the supernatant was loaded on a Strep-TactinXT Superflow cartridge (Iba) using an ÄKTA pure chromatography system. After washing the protein was eluted in two consecutive steps for 30 min with PBS containing 50 mM Biotin. Protein purity was confirmed via SDS PAGE.

## Western blotting

Cells were lysed in lysis buffer and protease inhibitor cocktail. Cell lysates were separated on 4–15% Criterion TGX Stain-Free protein gels (5678085, Bio-Rad), transferred to nitrocellulose membranes (0.2 μm, 10600001, Amersham) and blocked in 5% milk in PBS containing 0.1% Tween-20 5% milk. Membranes were incubated overnight at 4°C with a 1:1000 dilution of the following primary antibodies: StrepMAB (2-1507-001, Iba), SLC7A11 (12691S, Cell Signaling Technology, CST), HO-1 (ADI-SPA-895, Enzo), Vinculin (13901, CST), and Grb2 (610112, BD). Membranes were washed and incubated with 1:10,000 HRP-conjugated secondary antibodies Goat Anti-Rabbit and Goat Anti-Mouse (respectively, 111-035-003 and 115-035-003, Jackson ImmunoResearch) before visualization with SuperSignal West Pico Substrate (34080; Pierce).

## Cell death analysis

Cells were seeded in 48-well plates and, 24 hr after plating, ferroptosis was induced with either 10 μM erastin or 1 μM RSL3. Cells were monitored by live phase-contrast microscopy using the Incu-Cyte S3 system with the 10X objective, taking nine images per replicate for 36–48 hr as shown in the figures. Cell death was monitored over time using CellTox Green.

## Cell death analysis by flow cytometry

Ferroptosis was induced using ferroptosis inducers (FINs): erastin, RSL3, FINO2 (provided by Keith Worpel and Brent Stockwell), and ferroptocide (provided by Paul Hergenrother; *Llabani et al., 2019*). Cells were seeded in 6-well plates and, 24 hr after plating, treated with the indicated compounds. We used 5 μM erastin, 1.13 μM RSL3, 10 μM FINO2, and 10 μM ferroptocide. As a control for protection from ferroptosis, we used the ferrostatin Fer-1 at 1 μM. At indicated time points, cells were harvested, washed twice in PBS and stained with 5 μl of 7-AAD (BD Biosciences) and 5 μl of annexin-V-FITC (BD Biosciences) in 100 μl annexin-V binding buffer (BD Biosciences). After 15 min, cells were recorded on the LSRII with the FACS Diva 6.1.1 software (BD Biosciences).

## Quantification of lipid peroxidation

Lipid peroxidation was assessed using C11-BODIPY 581/591 (Molecular Probes/Life Technologies) according to the manufacturer's instructions.

## DPPH scavenging assay

Radical scavenging activity of PP, 4HPP, and I3P was analyzed using the stable free radical 1,1-diphenyl-2-picrylhydrazyl (DPPH). Fer-1 and ascorbic acid were used as positive controls. All compounds were used at a concentration of 200 μM and added to a 200 μM DPPH solution in pure methanol. After 10 min incubation, the absorbance was measured at 517 nm. The decrease in absorbance due to radical scavenging was calculated relative to the $H_2O$ control.

## GSH/GSSG assay

Quantification of GSH/GSSG ratio in the cell lysates of HeLa cells treated for 24 hr with 200 μM I3P was performed as previously described (*Rahman et al., 2006*).

## I3P fluorescence spectrum

I3P was dissolved at a 2 mM in ddH$_2$O. The fluorescence was measured in a black microplate using a CLARIOstar Plus microplate reader (BMG Labtech). For the excitation spectrum, the fluorescence intensity was measured for excitation wavelengths from 320 to 440 nm with a stepwidth of 1 nm at an emission wavelength of 470 nm (20 nm bandwidth). The emission spectrum was generated using an excitation wavelength of 395 nm (20 nm bandwidth) and detecting the emission at wavelengths from 430 to 600 nm with a stepwidth of 1 nm.

## I3P uptake

To monitor I3P uptake, HeLa cells ($2 \times 10^4$) were seeded in triplicates into 48-well plates and treated with 200 µM of Indole-3-pyruvic acid dissolved in the culture medium. AmCyan fluorescence was acquired at 2, 6, and 24 hr on a LSRFortessa with DIVA software (BD Biosciences). All flow data were analyzed by FlowJo software (version 10.5.0, Tree Star Inc, Ashland, OR).

## LAAO enzymatic assay

Specific LAAO activity toward the proteinogenic amino acids was measured by determination of H$_2$O$_2$ generation using Amplex UltraRed (AUR). Recombinant *N. naja* LAAO, human or murine IL4i1 were added to a mixture containing 1 U/ml HRP, 50 µM AUR, and 1 mM of the specific L-amino acid (CELLPURE, Roth) in 50 mM Sodium Phosphate (Sigma), pH 7.0 in a final volume of 100 µl per well of a black 96-well microplate. H$_2$O$_2$ production was measured over 5 min at 37°C using a CLARIOstar Plus microplate reader (BMG Labtech) at 530 nm excitation and 590 nm emission wavelength.

## Untargeted metabolomics

DMEM containing 10% FBS was incubated for 24 and 72 hr with 1 µg/ml of recombinant human IL4i1. The identical medium without IL4i1 was used as control. All samples were prepared in triplicates and frozen in liquid nitrogen. For analysis, samples were thawed on ice and proteins were precipitated with methanol. Upon centrifugation at 4°C for 10 min at 15,000 rcf, the extracts were transferred to a new tube. 10 µl of each sample were pooled and used as a quality control (QC) sample and the extracts were randomly assigned into the autosampler. Metabolites were separated on an iHILIC-(P) Classic HPLC column (HILICON, 100 × 2.1 mm; 5 µm; 200 Å; equipped with a guard column) with a flow rate of 100 µl/min delivered through an Ultimate 3000 HPLC system (Thermo Fisher Scientific). The stepwise gradient of HILIC analysis (*Wernisch and Pennathur, 2016*) involved starting conditions of 90% A (100% ACN), ramp to 25% B (25 mM ammonium hydrogen carbonate) within 6 min, 2 min hold at 25% B, ramp to 60% B from 8 to 21 min, switch to 80% B at 21.5 min, following by a flushing (21.5–26 min: 80% B) and re-equilibration step (26.1–35 at 10% B). All samples were analyzed by HILIC separation followed by ESI-HRMS in polarity switching mode in the mass range of 70–900 m/z. Sample spectra were acquired by data-dependent high-resolution tandem mass spectrometry on a Q Exactive Focus (Thermo Fisher Scientific). Ionization potential was set to +3.5/−3.0 kV, the sheath gas flow was set to 20, and an auxiliary gas flow of 5 was used. Samples were analyzed in a randomized fashion bracketed by a blank (MeOH: H$_2$O; 2:1 v/v) and QC sample for background correction and normalization of the data, respectively. QC samples were additionally measured in confirmation and discovery mode to obtain further MS/MS spectra for identification. The obtained data sets have been processed by compound discoverer (CD) 3.1.0.035 (Thermo Fisher Scientific). Our internal database, mzCloud database, and ChemSpider database (BioCyc, HMDB, KEGG) were used for compound annotation. A mass accuracy of 5 ppm was set and for our internal database a retention time window of 0.4 min was considered to compare against authentic standards. Acetonitrile (ACN) HiPerSolv CHROMANORM for HPLC-Supergradient was obtained from VWR Chemicals, methanol Optima LC/MS Grade from Fisher Chemicals, formic acid Suprapur was purchased from Merck, and H$_2$O was obtained from a Mili-Q Advantage A10 water purification system (Merck).

## RNA sequencing and analysis

THP-1 cells were treated in triplicates for 24 hr with 200 µM of PP, 4HPP, or I3P dissolved in RPMI medium. RPMI medium was used as control. HeLa cells were treated in triplicates for 24 hr with

DMEM containing 200 µM I3P or DMEM pre-incubated with 1 µg/ml murine IL4i1 (WT or K351A mutant). DMEM medium was used as control.

RNA was extracted using Qiagen RNeasy Mini kits (Qiagen) according to the manufacturer's protocol. mRNA sequencing libraries were prepared with 1 µg of total RNA of each sample using the NEBNext Ultra II Directional RNA Library Prep Kit for Illumina (E7765, NEB) with NEBNext Poly(A) mRNA Magnetic Isolation Module (E7490, NEB), according to standard manufacturer's protocol. Total RNA and the final library quality controls were performed using Qubit Flex Fluorometer (Q33327, Thermo Fisher Scientific) and 2100 Bioanalyzer Instrument (G2939BA, Agilent) before and after library preparation. Paired-end sequencing was performed on Illumina NextSeq 500 (2 × 43 bp reads). The samples were multiplexed and sequenced on one High Output Kit v2.5 to reduce a batch effect. BCL raw data were converted to FASTQ data and demultiplexed by bcl2fastq Conversion Software (Illumina). BAM and bigwig files are generated by STAR alignment and file conversion scripts – bam2wig and wigToBigWig.

After quality checking using the tool FastQC (v.0.11.7) the files were mapped to the human genome (Genome build GRCh38) downloaded from Ensembl using the star aligner (v. 2.7.3a) (*Dobin et al., 2013*). The mapped files were then quantified on a gene level based on the Ensembl annotations, using the featureCounts (*Liao et al., 2014*) tool from the SubRead package (*Liao et al., 2013*) (v. 1.4.6-p4).

Using the DESeq2 package (R 3.6.0, DESeq version 1.26.0) (*Anders and Huber, 2010*) the count data was normalized by the size factor to estimate the effective library size. After calculating the gene dispersion across all samples, the comparison of each two different conditions resulted in a list of differentially expressed genes for each comparison. A filtering step of removing genes with no reads in at least three samples was used.

Genes with an adjusted p-value <0.05 were then considered to be differentially expressed for generation of volcano plots. Gene ontology (GO) biological process overrepresentation analysis was computed with the web-based PANTHER tool using Fisher's exact test and false discovery rate (FDR) calculation. GO term overrepresentation was analyzed for the most significantly I3P-upregulated genes with a p-value cutoff of $p<10^{-9}$ using all detected genes from the dataset as the reference.

Clustering analysis was performed using the Perseus software (version 1.6.14.0). Genes were filtered for protein-coding genes and the reads log2 transformed. Multiple sample testing using ANOVA with an FDR threshold of 0.0001 was conducted and z-scores generated for the significant genes, which were used for hierarchical clustering (Euclidean distance, average linkage) generating a heat map.

## Statistical analyses

Statistical analyses were performed using GraphPad Prism software program (version 7, http://www.graphpad.com). Statistical tests are reported in the figure legends.

## Acknowledgements

We thank Matthias Feige and Karen Hildenbrand for discussion and experimental insights and Rin Ho Kim and Assa Yeroslaviz for RNA sequencing and bioinformatic analysis of the RNAseq data. We thank Alexander Strasser for technical support; Claudia Strasser, Leopold Urich, and Judith Scholz for protein purification.

Work in the PJM laboratory is supported by FOR-2599 (type 2 tissue immunity), SFB-TRR 127 from the Deutsche Forschungsgemeinschaft (DFG) and the Max-Planck-Gesellschaft. Work in the AL laboratory is supported by the SFB-TRR 205, SFB-TRR 127, the international research training group (IRTG) 2251 and by the DFG (Heisenberg-Professorship to AL [project number 324141047]). The CO lab is supported by grants from the European Research Council (ERC starting grant project number 716718), the DFG project number 395357507, SFB-1371, and grant number OH 282/1-1 within FOR-2599. The VBCF Metabolomics Facility is supported by the City of Vienna through the Vienna Business Agency.

## Additional information

### Competing interests

Thomas Köcher: Thomas Köcher is affiliated with Vienna BioCenter Core Facilities GmbH. The author has no financial interests to declare. The other authors declare that no competing interests exist.

### Funding

| Funder | Grant reference number | Author |
|---|---|---|
| Deutsche Forschungsge-meinschaft | SFB-TRR 127 | Andreas Linkermann Peter J Murray |
| Deutsche Forschungsge-meinschaft | FOR 2599 | Caspar Ohnmacht Peter J Murray |
| Deutsche Forschungsge-meinschaft | SFB-TRR 205 | Andreas Linkermann |
| Max-Planck-Gesellschaft | | Peter J Murray |
| International Research Training Group | 2251 | Andreas Linkermann |
| Deutsche Forschungsge-meinschaft | 324141047 | Andreas Linkermann |
| European Research Council | 716718 | Andreas Linkermann |
| Deutsche Forschungsge-meinschaft | OH 282/1-1 | Andreas Linkermann |

The funders had no role in study design, data collection and interpretation, or the decision to submit the work for publication.

### Author contributions

Leonie Zeitler, Conceptualization, Formal analysis, Validation, Investigation; Alessandra Fiore, Claudia Meyer, Marion Russier, Gaia Zanella, Formal analysis, Validation, Investigation; Sabine Suppmann, Formal analysis, Methodology; Marco Gargaro, Conceptualization, Resources, Formal analysis; Sachdev S Sidhu, Somasekar Seshagiri, Caspar Ohnmacht, Francesca Fallarino, Conceptualization, Resources; Thomas Köcher, Resources, Formal analysis; Andreas Linkermann, Conceptualization, Formal analysis, Supervision; Peter J Murray, Formal analysis, Supervision, Funding acquisition, Validation, Investigation, Writing - original draft, Writing - review and editing

### Author ORCIDs

Alessandra Fiore (iD) https://orcid.org/0000-0003-2070-6046
Marion Russier (iD) https://orcid.org/0000-0002-9633-9804
Sachdev S Sidhu (iD) http://orcid.org/0000-0001-7755-5918
Peter J Murray (iD) https://orcid.org/0000-0001-6329-9802

### Decision letter and Author response

Decision letter https://doi.org/10.7554/eLife.64806.sa1
Author response https://doi.org/10.7554/eLife.64806.sa2

## Additional files

### Supplementary files

- Supplementary file 1. I3P RNAseq GO terms.

- Transparent reporting form

## Data availability

RNAseq data have been deposited in GEO under accession codes GSE161159 and GSE167136.

The following datasets were generated:

| Author(s) | Year | Dataset title | Dataset URL | Database and Identifier |
|---|---|---|---|---|
| Zeitler L | 2021 | Analysis of THP-1 cell transcriptome changes induced by phenylpyruvate, 4-hydroxyphenylpyruvate and indole-3-pyruvate | https://www.ncbi.nlm.nih.gov/geo/query/acc.cgi?acc=GSE161159 | NCBI Gene Expression Omnibus, GSE161159 |
| Zeitler L | 2021 | Analysis of HeLa cell transcriptome changes induced by indole-3-pyruvate and mIL4i1 | https://www.ncbi.nlm.nih.gov/geo/query/acc.cgi?acc=GSE167136 | NCBI Gene Expression Omnibus, GSE167136 |

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
