## [Decision Letter]

**Acceptance summary:**

This paper identifies and characterizes an unexpected cytoprotective effect for the human L-amino acid oxidase, interleukin-4-induced-1 (IL4i1). Unlike the related snake venom L-amino acid oxidase which robustly generates hydrogen peroxide to induce ferroptotic cell death, the human protein has slower/attenuated activity and produces metabolites that protect cells from ferroptosis, via direct free radical scavenging activity and the transcriptional induction of an anti-ferroptotic cellular response. The study is elegant and the findings are exciting and of broad relevance to ferroptosis and cellular stress responses.

**Decision letter after peer review:**

Thank you for submitting your article "Anti-ferroptotic mechanism of IL4i1-mediated amino acid metabolism" for consideration by *eLife*. Your article has been reviewed by three peer reviewers, one of whom is a member of our Board of Reviewing Editors, and the evaluation has been overseen by Carla Rothlin as the Senior Editor. The reviewers have opted to remain anonymous.

The reviewers have discussed the reviews with one another and the Reviewing Editor has drafted this decision to help you prepare a revised submission.

We would like to draw your attention to changes in our policy on revisions we have made in response to COVID-19 (https://elifesciences.org/articles/57162). Specifically, when editors judge that a submitted work as a whole belongs in *eLife* but that some conclusions require a modest amount of additional new data, as they do with your paper, we are asking that the manuscript be revised to either limit claims to those supported by data in hand, or to explicitly state that the relevant conclusions require additional supporting data.

Summary:

This is a very interesting story showing that IL4iL, a secreted amino acid oxidase, has anti-ferroptotic properties. Unlike snake venom amino acid oxidases which robustly generate hydrogen peroxide to induce cell death, IL4iL products, including most notably I3P, protect against ferroptosis. IL4iL products have direct free radical scavenging activity and also elicit the induction of an anti-ferroptotic cellular response, in part via induction of HO-1. The authors also suggest that these effects of IL4iL may contribute to its pro-tumoral actions.

Essential revisions:

1) Can the authors do a genetic experiment to support or refute the protective role of HO-1 induction by I3P, or perhaps attenuate their comments that HO-1 contributes to the protective effects?

2) Is IL4i1 cytoprotective against snake venom LAOO?

3) The authors should show their RNAseq data for IL4i1, to enable comparison with the RNA-seq data for I3P.

Reviewer #1:

This is a very interesting story showing that IL4iL, a secreted amino acid oxidase, has anti-ferroptotic properties. Unlike snake venom amino acid oxidases which robustly generate hydrogen peroxide to induce cell death, IL4iL products, including most notably I3P, protect against ferroptosis. IL4iL products have direct free radical scavenging activity and also elicit the induction of an anti-ferroptotic cellular response, in part via induction of HO-1. The authors also suggest that these effects of IL4iL may contribute to its pro-tumoral actions.

This is a relatively complete story. Below are a few points for the authors' considerations.

1) The authors could consider using HO-1 knockdown to complement their findings with ZnPP in Figure 5.

2) Can the authors speculate on why there seems to be different mechanisms of action of I3P against RSL3 and erastin induced ferroptosis?

3) Can the authors comment more on the expression patterns of IL4iL, other than cancer and a few cell types?

Reviewer #2:

The authors may want to consider a scenario in which the genes encoding the snake venom LAAO and IL4i1 co-evolved such that one elicits and the other suppresses ferroptosis, respectively. This would be consistent with both proteins having the same core enzymatic activity, with LAAO inducing sufficient levels of H_2_O_2_ to kill cells and IL4i1 induces lower levels of H_2_O_2_, not sufficient to kill cells but likely sufficient to trigger a cytoprotective response against higher levels of H_2_O_2_. This would be reminiscent of other "desensitisation systems", where exposure to a sub-lethal dose of a given agonist renders cells, tissues, organs or organisms refractory to a subsequent dosage of the same substance that would otherwise be deleterious, i.e. cytotoxic. These adaptive responses are sometimes referred to as hormesis or accommodation and are based on the activation of evolutionarily conserved cytoprotective programs, similar to the ones put forward by the authors.Related to Figure 1:

• Is the snake venom LAAO internalised by mammalian cells?

• Does snake venom LAAO triggers the generation/accumulation of intracellular H_2_O_2_? This could be assessed, for example, using specific intracellular H_2_O_2_ probes.

• Does snake venom LAAO induces ferroptosis? This could be assessed, for example, using different pharmacologic ferroptosis inhibitors.

• It is intriguing that when mammalian cells express the gene encoding the snake venom LAAO they do not "succumb to their own poison". Can the authors discuss why this maybe.

• In the assays shown in F-H, please indicate what the error bars stand for (STD?, SEM?) and whether these reflect variation inherent to technical replicates (N=3) in a representative assay or from averaged data in independent assays. This comment applies to other figures using the same approach.

Related to Figure 2:

Similar to the previous comments related to Figure 1:

• Is IL4i1 internalised by mammalian cells?

• Does IL4i1 triggers the generation of intracellular H_2_O_2_. This could be assessed, for example, using specific intracellular H_2_O_2_ probes.

• Is IL4i1 cytoprotective against snake venom LAAO?

• There is no information on the experimental variation for the data illustrated in Figure 2F.

• Figure 2F. 1µg/ml of IL4i1 is used to treat the cells. Information on the plasma concentrations of IL4i1 in human or mouse under steady state and inflammatory conditions would be supportive to demonstrate the physiological function of IL4i1.

Related to Figure 3:

The authors investigated if the metabolic products of the IL4i1 enzymatic reaction modulate the "cell state", by performing RNAseq analysis on human THP-1 monocytic cells treated with I3P, 4HPP or PP. For comparison the authors should illustrate their RNAseq data (same format as in A) in cells exposed to IL4i1, further demonstrating the overlap between the profiles of gene expression obtained with I3P and IL4i1.

Related to Figure 5:

In Figure 5D, E, cellular cysteine, glutamate content and GSH/GSSG ratio are needed to support the conclusions reached.

Related to Figure 6:

• Although there is evidence for the anti-ferroptotic effect of Slc7a11 and Hmox1, the author did not show whether Slc7a11 is required for the protective effect of I3P, which is important to support the conclusions reached.

• Figure 6—figure supplement 1D, wash-out experiment is needed to exclude the scavenging effect of filtered medium.

In the assays shown in F-H, please indicate what the error bars stand for (STD?, SEM?) and whether these reflect variation inherent to technical replicates (N=3) in a representative assay or where these relate to averaged data from independent assays. This comment applies to other figures using the same approach.

Reviewer #3:

Congratulations on an exciting paper. I would suggest trying to determine the amino acid concentrations in the cell culture medium you used, and testing the effects of these two enzymes in medium containing natural concentrations of amino acids (the concentrations found in human plasma for example).

I would also suggest a genetic experiment to support or refute the protective role of HO-1 induction by I3P. I suspect HO-1 does not contribute to the protective effect and that ZnPP suppresses the protective effect of I3P via other mechanisms. Also, I believe that many of the genes induced by I3P in Figure 3C are targets of ATF4, such that knockdown of ATF4 should be informative about the protective effect of many of these.

Finally, good controls for I3P's protective effects would be tryptophan and indole, as they would have the same indole moiety. I would suggest comparing these side by side.

---

## [Author Response]

Essential revisions:1) Can the authors do a genetic experiment to support or refute the protective role of HO-1 induction by I3P, or perhaps attenuate their comments that HO-1 contributes to the protective effects?

In the original submission, we observed that the HO-1 inhibitor, ZnPP reversed the anti-ferroptotic effects of I3P. We also used ketoconazol, another HO-1 inhibitor with a distinct chemical structure from ZnPP. Like ZnPP, ketoconazol significantly reduced the protective effect of I3P from erastin-induced ferroptosis (Figure 5—figure supplement 2B, C). Collectively, these data argue that a heme oxygenase plays a key role in the anti-ferroptotic pathway elicited by I3P but did not answer the question of a direct role of HO-1. Next, we performed a siRNA-mediated knockdown of HO-1 to evaluate the specific contribution of HO-1. HO-1 knockdown, which was substantial as validated by immunoblotting (Figure 5—figure supplement 2D), did not interfere with the protective effect of I3P (Figure 5—figure supplement 2E). Thus, we concluded that effect the two HO-1 inhibitors is either (i) mediated by off-target effects of the compounds, (ii) inhibition of HO-2, which is highly expressed at the transcript level in HeLa cells (from our RNAseq data), or (iii) other heme-dependent enzymes (or, of course, combinations of the above). I3P-induced anti-oxidant gene expression includes a complex network and a simultaneous targeting of multiple proteins is necessary to elucidate the protection mechanism. We have therefore modified the description and outcomes of these experiments (Results).

2) Is IL4i1 cytoprotective against snake venom LAOO?

We did not observe the protection of Hela cells from the venom LAAO by medium pre-treated for 72 hr with IL4i1 (Figure 6—figure supplement 1F). Further, the ferroptosis inhibitors ferrostatin-1 and liproxstatin-1 did not protect from LAAO-induced cell death (Figure 6—figure supplement 1G), indicating that the snake venom LAAO does not induce ferroptosis. Thus, mammalian IL4i1 protects cells from ferroptotic cell death but not from the H_2_O_2_-dependent cell death caused by the snake venom LAAO, consistent with the differing enzyme kinetics and substrate range we articulated in the initial part of the Results section. We modified the manuscript to incorporate this information (Results).

3) The authors should show their RNAseq data for IL4i1, to enable comparison with the RNA-seq data for I3P.

In response to this comment, we performed a new RNAseq experiment to compare gene expression in HeLa cells incubated in DMEM containing I3P compared with DMEM pre-incubated for 72 hr with either WT or K351A mutant IL4i1. The outcomes of this experiment were remarkably clear and informative. We found almost identical gene expression patterns of untreated cells and cells treated with the K351A mutant IL4i1, while the I3P-treated cells clustered with the WT IL4i1-treated cells (Figure 6A). Also, we found a substantial overlap of the most significantly upregulated genes by I3P and WT IL4i1 treatment (Figure 6B), suggesting that I3P generation contributes to the gene expression changes mediated by IL4i1. These data are described in the Results and added to Figure 6. This experiment substantially extended the conclusions gained from the initial RNAseq data in the first submission.

Reviewer #1:This is a very interesting story showing that IL4iL, a secreted amino acid oxidase, has anti-ferroptotic properties. Unlike snake venom amino acid oxidases which robustly generate hydrogen peroxide to induce cell death, IL4iL products, including most notably I3P, protect against ferroptosis. IL4iL products have direct free radical scavenging activity and also elicit the induction of an anti-ferroptotic cellular response, in part via induction of HO-1. The authors also suggest that these effects of IL4iL may contribute to its pro-tumoral actions.This is a relatively complete story. Below are a few points for the authors' considerations.1) The authors could consider using HO-1 knockdown to complement their findings with ZnPP in Figure 5.

We addressed this above.

2) Can the authors speculate on why there seems to be different mechanisms of action of I3P against RSL3 and erastin induced ferroptosis?

We speculate that the requirement for radical scavenging to protect from RSL3 (Figure 5E) is best explained by RSL3 directly targeting GPX4, the enzyme catalyzing the key step in the cellular anti-ferroptosis mechanism(s) (Figure 4A) by detoxifying lipid peroxides. Thus, gene expression increasing anti-oxidative pathways upstream of GPX4, e.g. increase of cysteine import by increased levels of SLC7A11 (Figure 5G) or increased generation of GSH (Figure 5F), do not have protecting effects once GPX4 is blocked. Nevertheless, we do not want to exclude that the scavenging effect could also contribute to the protection from erastin and we clarified this concept in the main text (Results).

3) Can the authors comment more on the expression patterns of IL4iL, other than cancer and a few cell types?

At this point in the evolution of research into IL4i1, we are limited by reagents to directly answer, or even speculate on this issue beyond scRNAseq data. For example, a specific antibody to the mouse IL4i1 for IHC or immunoblotting is not yet available (in unpublished studies, we have tested all commercially available anti-IL4i1 antibodies for mouse and found them to be non-specific). So far, a reporter allele has not yet been described. Nevertheless, our scRNAseq experiments related to other projects indicate that IL4i1 is expressed only in immune cells, and predominantly in myeloid cells. Our conclusions (so far) do not differ from the current literature that we cite in the manuscript. In the future, we hope a highly specific and knockout validated anti-IL4i1 antibody will accelerate research in mouse models, and that reporters will allow us to track IL4i1 in, for example, infection or cancer models.

Reviewer #2:The authors may want to consider a scenario in which the genes encoding the snake venom LAAO and IL4i1 co-evolved such that one elicits and the other suppresses ferroptosis, respectively. This would be consistent with both proteins having the same core enzymatic activity, with LAAO inducing sufficient levels of H_2_O_2_ to kill cells and IL4i1 induces lower levels of H_2_O_2_, not sufficient to kill cells but likely sufficient to trigger a cytoprotective response against higher levels of H_2_O_2_. This would be reminiscent of other "desensitisation systems", where exposure to a sub-lethal dose of a given agonist renders cells, tissues, organs or organisms refractory to a subsequent dosage of the same substance that would otherwise be deleterious, i.e. cytotoxic. These adaptive responses are sometimes referred to as hormesis or accommodation and are based on the activation of evolutionarily conserved cytoprotective programs, similar to the ones put forward by the authors.Related to Figure 1:• Is the snake venom LAAO internalised by mammalian cells?

To address this question we performed immunoblot analysis of HeLa cells treated for 2 and 4 hr with 1 µg/ml of the recombinant LAAO enzyme. As shown in Author response image 1, we could not detect any specific bands for the protein in the cell lysates, also not after overexposure of the part of the membrane with the cell lysates (data not shown). This suggests that the enzyme is probably not taken up by the cells. Of course, this experiment does not exclude the possibility that IL4i1 is taken up and immediately degraded. Nevertheless, within the limit of the assay, we cannot detect any “cell associated” IL4i1. So far, there is no evidence for an IL4i1 “receptor” and indeed we have little reason to consider such a mechanism is plausible, as all our data using enzyme-dead mutants reported in our manuscript point to IL4i1 functioning as an extracellular amino acid-metabolizing enzyme.

**Author response image 1. sa2fig1:** (A) Immunoblotting of LAAO in HeLa cell lysates and supernatant after 2 and 4 hr treatment with 1µg/ml of the recombinant LAAO enzyme. (B) ROS quantification in HeLa cells treated for 4 hr with WT and mutant LAAO by flow cytometry using the ROS probe H_2_DCFDA. 200 µM H_2_O_2_ was used as a positive control. n=3 biological replicates, the graph is representative for 3 independent experiments. Data was analyzed by one-way ANOVA with Tukey's multiple comparisons test; *p<0.05; ***p<0.001; ns = not significant. (C) Immunoblotting of IL4i1 in HeLa cell lysates and supernatant after 4 and 24 hr treatment with 1µg/ml of the recombinant mouse and human IL4i1 enzyme. (D) ROS quantification in HeLa cells treated for 4 or 24 hr with human, mouse WT and mouse mutant IL4i1 by flow cytometry using the ROS probe H_2_DCFDA. n=3 biological replicates, the graph is representative of 3 independent experiments. Data was analyzed by one-way ANOVA with Tukey's multiple comparisons test; ns = not significant. (E) HeLa cells were concurrently treated with erastin or RSL3 in presence of 200 µM I3P, 200 µM Indole or 200 µM L-Trp. n=3 biological replicates, the graph is representative for 3 independent experiments. All error bars represent standard deviation.

• Does snake venom LAAO triggers the generation/accumulation of intracellular H_2_O_2_? This could be assessed, for example, using specific intracellular H_2_O_2_ probes.

To detect intracellular H_2_O_2_, we performed flow cytometry using the ROS probe H_2_DCFDA dye. However, since extracellular H_2_O_2_ can be taken up into cells (Bienert et al., 2006) it is not possible to conclusively distinguish between intra- or extracellular generated peroxide. However, 4 hr after treatment with WT LAAO we could observe an increased H_2_DCFDA staining as compared to cells treated with the enzyme-dead version (Author response image 1). In this experiment 200 µM H_2_O_2_ was used as a positive control.

• Does snake venom LAAO induces ferroptosis? This could be assessed, for example, using different pharmacologic ferroptosis inhibitors.

We addressed this issue in the major question part of the response above.

• It is intriguing that when mammalian cells express the gene encoding the snake venom LAAO they do not "succumb to their own poison". Can the authors discuss why this maybe.

We agree with the reviewer that this is a fascinating point and in part this question relates to the many “tricks” we need to purify the enzyme (detailed with more clarity in the Materials and methods section). The expression and yield of the snake venom LAAO is extremely low in comparison to the yield obtained from non-toxic proteins using the same expression system, suggesting that the expression is suppressed to a “non-toxic” level. This is why, for the LAAO purification, we first performed a concentration step from a large volume of media. We clarified this issue further in the Materials and methods section.

• In the assays shown in F-H, please indicate what the error bars stand for (STD?, SEM?) and whether these reflect variation inherent to technical replicates (N=3) in a representative assay or from averaged data in independent assays. This comment applies to other figures using the same approach.

The missing information in regard to error bars was corrected in each figure legend.

Related to Figure 2:Similar to the previous comments related to Figure 1:• Is IL4i1 internalised by mammalian cells?

To answer this question, which is similar to the question posed above, we performed Western Blot analysis of HeLa cells treated for 4 and 24 hr with 1 µg/ml of the recombinant mouse and human IL4i1. Since these proteins are not toxic we chose to also investigate a later time point than for the snake venom LAAO. We could not detect any specific bands for the proteins in the cell lysates (Author response image 1), also not after overexposure of the part of the membrane with the cell lysates (data not shown). This suggests that IL4i1 is probably not taken up by the cells, at least not in a range detectable by Western Blotting.

• Does IL4i1 triggers the generation of intracellular H_2_O_2_. This could be assessed, for example, using specific intracellular H_2_O_2_ probes.

As described above, we performed flow cytometry using H_2_DCFDA dye but we could not observe any significant changes in intracellular ROS levels caused by human or mouse IL4i1 (Author response image 1).

• Is IL4i1 cytoprotective against snake venom LAAO?

This issue was addressed above.

• There is no information on the experimental variation for the data illustrated in Figure 2F.

This information was missing is now corrected the figure and legend.

• Figure 2F. 1µg/ml of IL4i1 is used to treat the cells. Information on the plasma concentrations of IL4i1 in human or mouse under steady state and inflammatory conditions would be supportive to demonstrate the physiological function of IL4i1.

We thank the reviewer for highlighting this point; in our experiments we used 1µg/ml fixed concentration as a proof of concept for the generation of the metabolites. In later experiments (Figure 6C, D, F) we titrated the enzyme, to determine working concentration ranges for the IL4i1 protective effect. So far, there is not much data about serum levels of IL4i1, mainly due to the lack of specific detection antibodies (see the point above to reviewer 1 on IL4i1 expression). However, a recent study from Sadik et al., showed IL4i1 serum levels up to 300 pg/ml in a mouse leukemia model. Therefore, we can speculate that local IL4i1 tissue concentrations may be much higher. We added this to our Discussion.

Related to Figure 3:The authors investigated if the metabolic products of the IL4i1 enzymatic reaction modulate the "cell state", by performing RNAseq analysis on human THP-1 monocytic cells treated with I3P, 4HPP or PP. For comparison the authors should illustrate their RNAseq data (same format as in A) in cells exposed to IL4i1, further demonstrating the overlap between the profiles of gene expression obtained with I3P and IL4i1.

We addressed the overlap of the gene expression profiles between I3P and IL4i1 in the main of the response. We decided to perform this additional experiment in HeLa cells (major comment 3 above), since this cell line was used in all later experiments and to provide a further validation that the I3P-dependent activation of gene expression is not cell-dependent. The outcomes of this experimental series are shown in Figure 6A and B.

Related to Figure 5:In Figure 5D, E, cellular cysteine, glutamate content and GSH/GSSG ratio are needed to support the conclusions reached.

We agree that this is important information. We performed an assay to investigate whether I3P could modulate the intracellular GSH/GSSG ratio, because the GSH/GSSG ratio is the most “downstream” readout for these metabolites. As expected by the I3P-dependent SLC7A11 up-regulation, we found that I3P significantly increased the GSH/GSSG ratio in HeLa cells (Figure 5F and described in the Results).

Related to Figure 6:• Although there is evidence for the anti-ferroptotic effect of Slc7a11 and Hmox1, the author did not show whether Slc7a11 is required for the protective effect of I3P, which is important to support the conclusions reached.

We did not perform an experiment to show that SLC7A11 is contributing to the protective effect of I3P, because we are already targeting SLC7A11 itself by inducing ferroptosis with the SLC7A11 inhibitor erastin. This is a widely used approach in this field.

• Figure 6—figure supplement 1D, wash-out experiment is needed to exclude the scavenging effect of filtered medium.

We think that both, radical scavenging together with the activation of anti-oxidative networks mediate the protective effect in this setting as shown in Figure 5. Therefore, we do not want to exclude the contribution of the scavenging effect. Further experiments will be necessary to unequivocally separate the contributions of radical scavenging versus gene expression protective effects over time.

In the assays shown in F-H, please indicate what the error bars stand for (STD?, SEM?) and whether these reflect variation inherent to technical replicates (N=3) in a representative assay or where these relate to averaged data from independent assays. This comment applies to other figures using the same approach.Reviewer #3:Congratulations on an exciting paper. I would suggest trying to determine the amino acid concentrations in the cell culture medium you used, and testing the effects of these two enzymes in medium containing natural concentrations of amino acids (the concentrations found in human plasma for example).

To answer this question we created a medium containing human plasma concentrations of amino acids and glucose (Leney-Greene et al., 2020) and tested the ability of IL4i1 to create a ferroptosis-protective environment. We could confirm that WT IL4i1 also prevented ferroptosis under these medium conditions (Figure 6—figure supplement 1E). This is described in the Results.

I would also suggest a genetic experiment to support or refute the protective role of HO-1 induction by I3P. I suspect HO-1 does not contribute to the protective effect and that ZnPP suppresses the protective effect of I3P via other mechanisms.

We addressed this in the main part of the response.

Also, I believe that many of the genes induced by I3P in Figure 3C are targets of ATF4, such that knockdown of ATF4 should be informative about the protective effect of many of these.

We agree with the reviewer that interference with ATF4 may give important information on the exact pathways involved in I3P-mediated ferroptosis protection. We also think that the I3P-induced anti-oxidant gene expression includes the activation of AhR and Nrf2 pathways; the study of this complex network is beyond the purpose of this manuscript and will be a major objective of our future studies. We added this to the Discussion.

Finally, good controls for I3P's protective effects would be tryptophan and indole, as they would have the same indole moiety. I would suggest comparing these side by side.

We evaluated the potency of tryptophan and indole to interfere with erastin and RSL3 induced ferroptosis. We observed that tryptophan could not protect the cells whereas indole had an intermediate effect on cell protection both towards erastin- and RSL3- induced ferroptosis, however less than I3P (Author response image 1). We included the data on tryptophan in the manuscript (Figure 4—figure supplement 1E) to further confirm that the IL4i1-catalyzed conversion of tryptophan to I3P is required for the protective effect (Results).

References:

Bienert, G. P., J. K. Schjoerring and T. P. Jahn. 2006. Membrane transport of hydrogen peroxide. Biochim Biophys Acta 1758:994-1003. 10.1016/j.bbamem.2006.02.015, PMID: PMID